# Neuronal processing of noxious thermal stimuli mediated by dendritic Ca$^{2+}$ influx in *Drosophila* somatosensory neurons

**Shin-Ichiro Terada[1†], Daisuke Matsubara[1†], Koun Onodera[1†], Masanori Matsuzaki[2], Tadashi Uemura[1*], Tadao Usui[1*]**

[1]Graduate School of Biostudies, Kyoto University, Kyoto, Japan; [2]Division of Brain Circuits, National Institute for Basic Biology, Okazaki, Japan

**Abstract** Adequate responses to noxious stimuli causing tissue damages are essential for organismal survival. Class IV neurons in *Drosophila* larvae are polymodal nociceptors responsible for thermal, mechanical, and light sensation. Importantly, activation of Class IV provoked distinct avoidance behaviors, depending on the inputs. We found that noxious thermal stimuli, but not blue light stimulation, caused a unique pattern of Class IV, which were composed of pauses after high-frequency spike trains and a large Ca$^{2+}$ rise in the dendrite (the Ca$^{2+}$ transient). Both these responses depended on two TRPA channels and the L-type voltage-gated calcium channel (L-VGCC), showing that the thermosensation provokes Ca$^{2+}$ influx. The precipitous fluctuation of firing rate in Class IV neurons enhanced the robust heat avoidance. We hypothesize that the Ca$^{2+}$ influx can be a key signal encoding a specific modality.

**\*For correspondence:** tauemura@lif.kyoto-u.ac.jp (TU); tadao.usui@gmail.com (TU)

[†]These authors contributed equally to this work

**Competing interests:** The authors declare that no competing interests exist.

## Introduction

Animals can sense diverse sensory inputs such as optical stimuli, plumes of volatile chemicals, acoustic waves, and atmospheric temperature by using a variety of specific sensory organs. Among these, polymodal nociceptors are defined as sensory neurons that respond to aversive stimuli with distinct physical properties. For example, mammalian C-fiber nociceptors detect harsh touch, high temperature, acid stimulus, and a number of toxic chemicals(*Abraira and Ginty, 2013*; *Delmas et al., 2011*; *Perl, 1996*).

Ion channel proteins that are expressed in polymodal nociceptors of nematodes, insects, and mammals have been discovered, and nowadays, molecular basis of thermal or mechanical nociceptive sensory transduction has primarily converged on three protein superfamilies: the TRP channels, the PIEZO channels, and the DEG/ENaC channels, all of which are well-conserved across many species (*Basbaum et al., 2009*; *Geffeney and Goodman, 2012*; *Lumpkin et al., 2010*). TRP channels are nonspecific cation channels and several members of this family have been implicated in thermal nociception (*Lumpkin and Caterina, 2007*). For example, dTrpA1 is an essential component of both thermal and mechanical nociception in *Drosophila* (*Zhong et al., 2012*). PIEZO and DEG/ENaC channels are required specifically for the mechanical nociception (*Chatzigeorgiou et al., 2010*; *Coste et al., 2010*; *Kim et al., 2012b*).

Exposure to short-wavelength light is also one of the hazardous stimuli for terrestrial animals such as soil-dwelling nematodes and a number of boring insects including the larval forms of holometabolous insects (*Hori et al. 2014*). Both the *lite-1* gene in *C. elegans* and *Gustatory receptor 28b* (*Gr28b*) gene in *Drosophila* encode the light-activated G-protein-coupled receptors and are essential for the noxious light sensation eliciting stereotypical light-avoidance behaviors (*Liu et al., 2010*; *Xiang et al., 2010*; *Yamanaka et al., 2013*).

**eLife digest** Animals often need to get away quickly from dangers in their environment, such as temperatures that are hot enough to damage their tissues. As such, an animal's brain often encodes automatic 'avoidance responses' to signs of danger, which help the animal move away from harm.

The nervous system of a fruit fly larva, for example, contains a distinct class of neurons (known as class IV neurons) that respond specifically to high temperatures and ultraviolet or blue light. Both of these stimuli are potentially harmful, but the larvae escape from heat by rolling with a corkscrew-like motion, yet they turn their heads away from a source of ultraviolet or blue light. So, how does the same set of neurons orchestrate these two different types of behavior?

To answer this question, Terada, Matsubara, Onodera et al. measured the activity in the class IV neurons in two different ways. First, the levels of calcium ions in the neurons, which play a key role in neurons' activity, were imaged using a calcium-sensitive biosensor. Second, electrodes were used to directly on the class IV neurons to record changes in their electrical activity.

The experiments showed that class IV neurons responded to heat by producing a characteristic burst of electrical activity followed by a pause, and that this pattern of electrical activity was accompanied by a large rise in the calcium signal. In contrast, the same neurons did not show this 'burst and pause' pattern of activity when the fruit fly larvae were exposed to ultraviolet/blue light. Instead, these conditions triggered much smaller changes in electrical activity. Further experiments then confirmed that the characteristic 'burst and pause' pattern of electrical activity was linked to the rolling motion observed when the larvae try to escape from heat.

These findings show how differing patterns of activity in the same neurons can be used to differentiate between different types of stimuli. Further work is now needed to explain how these two different patterns of activity in one set of neurons is translated by the fruit fly's brain into different patterns of behavior.

In at least two noteworthy cases, organisms can react to different modes of stimuli through a single type of nociceptor: the *C. elegans* PVD neuron responds to a harsh poke and cold temperatures (*Chatzigeorgiou et al., 2010*), whereas one subclass of multidenritic neurons in the *Drosophila* larva perceives a sharp poke, noxious high temperatures, and UV or blue light (*Im and Galko, 2012*; *Xiang et al., 2010*). Intriguingly, *Drosophila* larvae exhibit distinct escape behavior outputs in response to the thermal or mechanical stimuli versus blue light (see below). To unveil the mechanism of this type of polymodal sensation and processing, we addressed how the distinct sensory inputs are transduced and encoded into differential firing patterns.

*Drosophila* dendritic arborization (da) neurons constitute a subfamily of sensory neurons that elaborate their dendritic arbors two-dimensionally on the basal surface of the epidermis (*Grueber et al., 2002*; *Han et al., 2012*; *Im and Galko, 2012*; *Jan and Jan, 2010*; *Kim et al., 2012a*; *Shimono et al., 2009*; *Yasunaga et al., 2010*). da neurons in the larval abdominal hemisegment are classified into four morphological categories, Classes I-IV, in order of increasing territory size and/or branching complexity (*Grueber et al., 2002*; *Jan and Jan, 2010*). A series of studies have shown that Class IV neurons are polymodal nociceptors and relevant receptors are identified as described below: Touching the dorsal cuticle of larvae with a metal probe hotter than 45°C evokes a stereotypic corkscrew-like escape rolling behavior (*Babcock et al., 2009*; *Tracey et al., 2003*); and activation of the Class IV neurons is necessary and sufficient for this output (*Hwang et al., 2007*; *Ohyama et al., 2013*). This thermal nociception behavior is mediated by thermoTRPs genes including *Drosophila* TRPA1 (dTrpA1), Painless, and Pyrexia (*Babcock et al., 2011*; *Hwang et al., 2012*; *Lee et al., 2005*; *Neely et al., 2010*; *2011*; *Tracey et al., 2003*; *Zhong et al., 2012*). In addition to the noxious heat, strong mechanical stimuli, such as a sharp poke, elicit the same nocifensive behavior (*Hwang et al., 2007*). Moreover, Class IV neurons are extra-ocular photoreceptors that respond to short-wavelength light stimuli and evoke a directional shift of locomotion, which requires Gr28b (*Berni et al., 2012*; *Xiang et al., 2010*; *Yamanaka et al., 2013*). Therefore, to achieve such a discriminative processing, Class IV neurons are supposed to integrate multiple modes of sensory inputs into distinct firing patterns in a highly sophisticated manner. Recently, extracellular single-unit

recordings have been developed to monitor physiological activities of Class I-IV dendritic arborization neurons upon naturalistic stimulation, such as light and gentle touch (*Tsubouchi et al., 2012*; *Xiang et al., 2010*; *Yan et al., 2013*; *Zhang et al., 2013*). However, there have been few extensive analyses of firing responses of Class I–IV neurons upon spatially restricted and temporally controlled noxious heat stimulations.

In this study, we built a new measurement system that combines three components: an infrared (IR) laser that allows quantitatively controlled step-like and local heating, neuronal calcium imaging, and extracellular single-unit recording to monitor firing patterns. In response to the local heating over 45°C, Class IV neurons generated unique firing patterns that were composed of intermittent pause periods after high-frequency firing trains. This 'burst and pause' firing pattern was associated with a large rise in $Ca^{2+}$ in the entire dendritic tree, which we designated as the $Ca^{2+}$ transient. We showed that both of the unique firing pattern and the $Ca^{2+}$ transient are physiological responses of Class IV neurons, characterized their underlying mechanisms by using pharmacological and genetic approaches, and compared the response to the noxious heat with that to short-wavelength light to address the basis of the polymodal sensation. Furthermore, we pursued which characteristics of the 'burst and pause' firing pattern contributes to the induction of the rolling behavior, and showed that the multiple fluctuations of firing rate enhanced robust output by using optogenetic manipulation. Altogether, we propose that the low-frequency continuous firing train transmits the noxious light sensation; in contrast, the multiple peaks of firing rate fluctuation facilitate the perception of a noxious heat stimulus.

## Results

### Infrared laser-mediated noxious thermal stimuli evoke high-frequency firing in Class IV neurons

To understand how Class IV neurons respond to noxious thermal stimuli, we built a new system employing a 1462-nm infrared (IR) laser as a step-like and local heating device (*Figure 1A*). The IR laser has been applied for the precise control of temperature with high spatial and temporal resolution, such as the biophysical analyses of the temperature dependency of thermoTRP channels (*Yao et al., 2009*) and heat-shock-mediated expression of transgenes in targeted single cells by IR-laser irradiation through a microscope objective (*Kamei et al., 2008*).

We first examined the in vitro heating profiles of the IR laser: the kinetics of temperature changes, controllability of heating, and spatial distributions of heat around the laser focus. To measure the microenvironmental temperature, we exploited the temperature dependence of the electrical resistance of a glass microelectrode (*Palmer and Williams, 1974*; *Shapiro et al., 2012*; *Yao et al., 2009*). The temperature within a radius of approximately 50 µm from the laser focus rapidly increased from an ambient temperature (25°C) to target temperatures (30–50°C) within 100 ms, and stayed almost constant at the maximum temperature throughout the 1 s irradiation (*Figure 1B and C*). Moreover, the temperature maximum was directly proportional to the power output of the laser (*Figure 1B*). Compared to the perfusion protocol involving a preheated solution that is introduced into the recording chamber (*Xiang et al., 2010*) as well as the direct heating of the chamber (*Liu et al., 2003*), the IR-laser system allowed us to reproducibly deliver more spatially restricted, and more temporally controlled stimuli.

Next, we stimulated Class IV neurons by means of this system and measured the cellular physiological responses. For this purpose, we dissected third instar *Drosophila* larvae, prepared fillet preparations, and performed extracellular single-unit recordings. The foci of the IR-laser irradiation were targeted onto the proximal dendritic arbors (the path length of branches between the IR-laser foci and soma was no longer than 100 µm). IR-laser irradiation of more than 30-mW laser output power evoked high-frequency firings in Class IV neurons (*Figure 1D and E*: 30 mW, 40.6 ± 4.8 Hz; 40 mW, 83.9 ± 7.3 Hz; $n$ = 9 cells). From our calibration of the IR laser, an irradiation of 1-mW raised the temperature by an increment of ~0.6°C in the focused region (inset of *Figure 1B*); thus, the estimated temperature was ~43°C for the 30-mW laser output power. This threshold temperature was close to that of previous experiments, where the jumps of the firing rate were detected at around 40°C with perfusion of the preheated solution into the recording chamber (*Xiang et al., 2010*). We were intrigued by the firing patterns in response to IR irradiation of 40-mW laser power, which typically

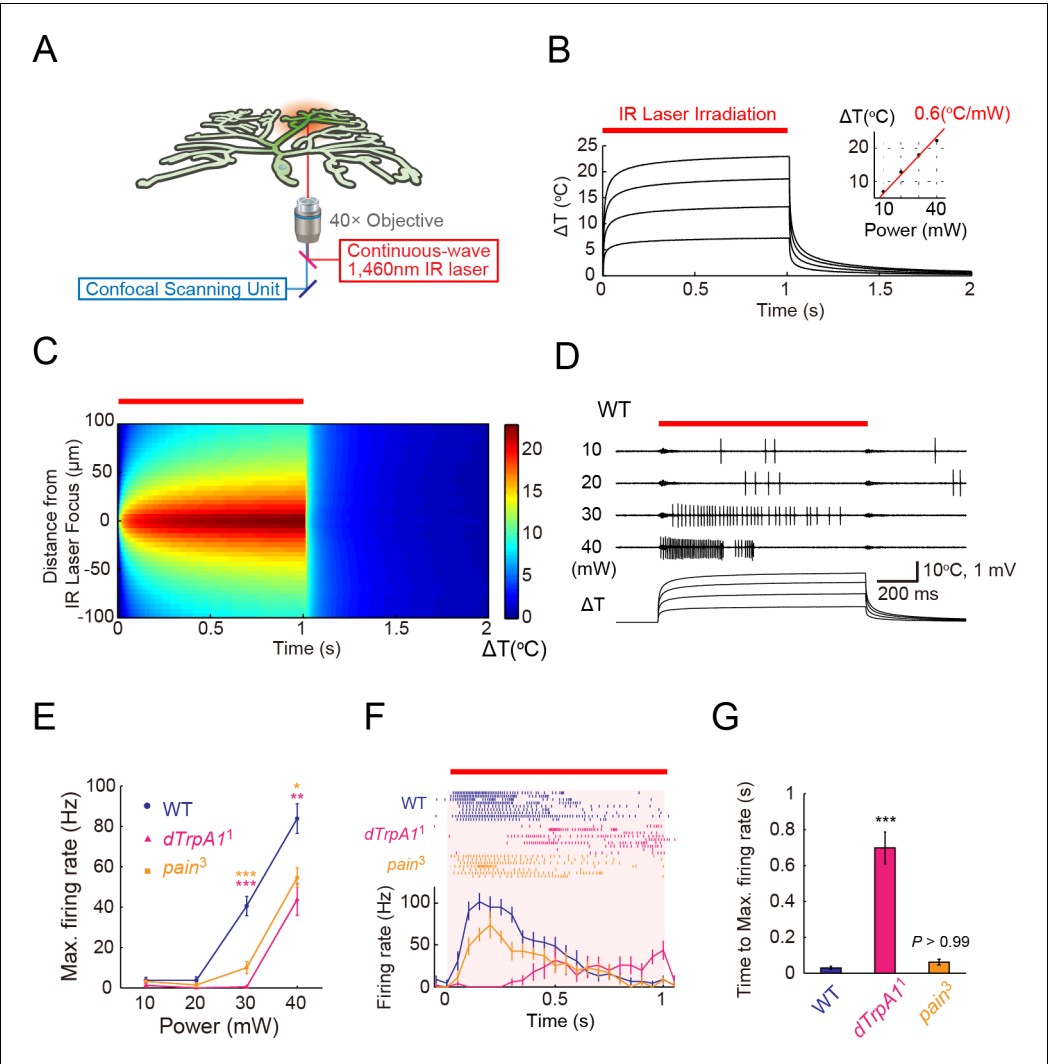

**Figure 1.** ThermoTRP-dependent responses of Class IV neurons to IR-laser irradiation. (**A**) A schematic diagram of the heat-stimulation system with the IR-laser irradiation. The IR-laser beam passed through an objective and targeted Class IV neurons in whole- or fillet-mounted larvae. The wavelength of IR matches the combination of symmetric and anti-symmetric OH stretching modes of water and can heat water efficiently (***Palmer and Williams, 1974***). We measured firing responses of the neuronal somata using an extracellular recording electrode in fillet preparations, and acquired ratiometric data of neuronal $Ca^{2+}$ fluctuations in whole-mount or fillet preparations. (**B** and **C**) Temporal and spatial profiles of the IR-laser induced temperature changes of the microenvironment. A specimen was irradiated with the IR-laser beam for 1 s (orange horizontal bar) and temperature changes at the laser focus were calculated on the basis of changes of electrical resistances of a solution in a glass microelectrode (see details in Methods). (**B**) Traces represent temporal changes of temperature at corresponding laser powers (10, 20, 30, and 40 mW). The inset shows a plot of temperature increments at 700 ms at each power, and the red line indicates a linear fit to the data (slope = +0.6°C/mW). (**C**) Pseudo-color map of spatiotemporal temperature changes around the laser focus (the zero point along Y-axis) in response to 40-mW laser irradiation and subsequent shut-off. The position of the glass microelectrode was shifted along the Y-axis at 5 µm intervals to monitor temporal changes at individual locations. At right is the color scale for the temperature change. (**D–G**) Extracellular single-unit recordings of Class IV neurons that were irradiated at various output powers (**D** and **E**) or at 40 mW (**F** and **G**) for 1 s (orange horizontal bar in **D** and **F**). A location in the proximal dendritic arbor was targeted. (**D**) Spike trains and time courses of temperature changes at IR foci (bottom). (**E**) Power-dependent increment in the maximum firing rate (defined in 'Materials and methods') of the wild type (circle), *dTrpA1*[1] mutant (triangle), and *pain*[3] mutant (square). (**F**) Raster plots of firing (top) and time course of firing rate (bottom) of the wild type and the mutants (bin size 50 ms). (**G**) Quantification of timing of maximum excitation ('Time to Max. Firing Rate' as defined in 'Materials and methods') of Class IV neuron of each genotype. Numbers of neurons

*Figure 1 continued on next page*

*Figure 1 continued*

tested were 9 (the wild type), 10 (*dTrpA1*[1]), and 7 (*pain*[3]), and data are presented as mean ± s.e.m. (E–G). * p < 0.05, ** p < 0.01, *** p < 0.001 versus wild type by unpaired *t*-test with Bonferroni correction (**E** and **G**). IR, infrared.

The following source data and figure supplements are available for figure 1:

**Source data 1.** The maximum firing rate of the wild type, *dTrpA1*[1] mutant, and *pain*[3] mutant (Columns A–F) and Quantification of timing of maximum excitation of Class IV neuron of each genotype (Columns H–J).

**Figure supplement 1.** ThermoTRP-dependent responses of Class IV neurons to IR-laser irradiation.

**Figure supplement 2.** Assessment of effects of IR-laser irradiation on Class IV neurons.

contained pauses that followed high-frequency spikes (see 40-mW in *Figure 1D*); and we characterized the underlying mechanistic details as described later. Our control or 'wild-type' larvae in the electrophysiological experiments carried a transgene of a $Ca^{2+}$ probe that was expressed selectively in Class IV neurons, unless described otherwise.

## Interplay of dTrpA1 and Painless generate putative thermocurrents in Class IV neurons

We investigated how the IR-laser irradiations evoked high-frequency firings in Class IV neurons by extracellular recording. We focused on the roles of two members of the TRPA channel family, dTrpA1 and Painless. Previous studies had shown that the activation of both of the two channels in the neurons was required to induce nocifensive escape behaviors (*Babcock et al., 2011*; *Zhong et al., 2012*), and that the expression of either of the channels was sufficient to elicit thermocurrents in cultured cell lines (*Sokabe et al., 2008*; *Wang et al., 2013*). Therefore, we expected that these molecules served as primary heat-activated cation channels in the neurons.

*dTrpA1* mutant (*dTrpA1*[1] homozygotes) and *pain* mutant (*pain*[3] homozygotes) exhibited normal basal firing in the neurons (data not shown), but showed significantly lower firing rates during the IR-laser irradiation (*Figure 1E*: [*dTrpA1*[1]] 30 mW: 0.5 ± 0.3 Hz, 40 mW: 43.8 ± 7.9 Hz, *n* = 10 cells; [*pain*[3]] 30 mW: 10.0 ± 3.0 Hz, 40 mW: 54.6 ± 4.8 Hz, *n* = 7 cells). Interestingly, *dTrpA1*[1] mutants displayed significantly longer latency time between the onset of the IR irradiation and the increase in firing rate when compared to the wild-type or *pain*[3] larvae (*Figure 1F and G*; Time to Max. Firing Rate in *Figure 1G*: [wild type] 45.0 ± 7.1 ms, [*dTrpA1*[1]] 698 ± 89.5 ms; [*pain*[3]] 61.6 ± 16.4 ms, p < $5.7 \times 10^{-6}$, *t*-test with Bonferroni correction; our results of other allelic combinations are described in *Figure 1—figure supplement 1*). The temporal pattern of the wild-type firing was not a simple summation of the two TRPA mutants; thus, we infer that the two heat-activated channels should function in a coordinated manner. Our results also strongly suggested that the firing responses of the wild-type neurons reflected intrinsic neuronal activities, dependent on the heat-activated channels and were not induced by photo damage (see also *Figure 1—figure supplement 2*).

## IR irradiation to the soma evokes dendritic $Ca^{2+}$ transients

Although the occurrences of intracellular $Ca^{2+}$ rises are often taken to be a hallmark of neural activity, it was not clear that the $Ca^{2+}$ rises were faithfully indicative of the activation of *Drosophila* sensory neurons. We therefore decided to confirm the functional relationship between the $Ca^{2+}$ rises and the neuronal firing responses. We first examined the dynamics of changes of cytoplasmic $Ca^{2+}$ concentration during IR-laser irradiations. For this purpose, transgenic strains were generated that highly and specifically expressed an intramolecular FRET-based $Ca^{2+}$ probe, TN-XXL (*Mank et al., 2008*), in Class IV neurons (see Methods and *Figure 2—figure supplement 1* for details). Using ratiometric imaging, we measured the $Ca^{2+}$ changes in the neurons in whole-mount larvae (*Figure 2A–E*) or fillet preparations (*Figure 2F*).

When the soma was irradiated with the IR laser, a large increase in $Ca^{2+}$ was observed over the entire dendritic tree, which we designated as dendritic $Ca^{2+}$ transients or simply $Ca^{2+}$ transients (*Figure 2A,B* and *Video 1*). The rise of the $Ca^{2+}$ transients emerged simultaneously in both proximal

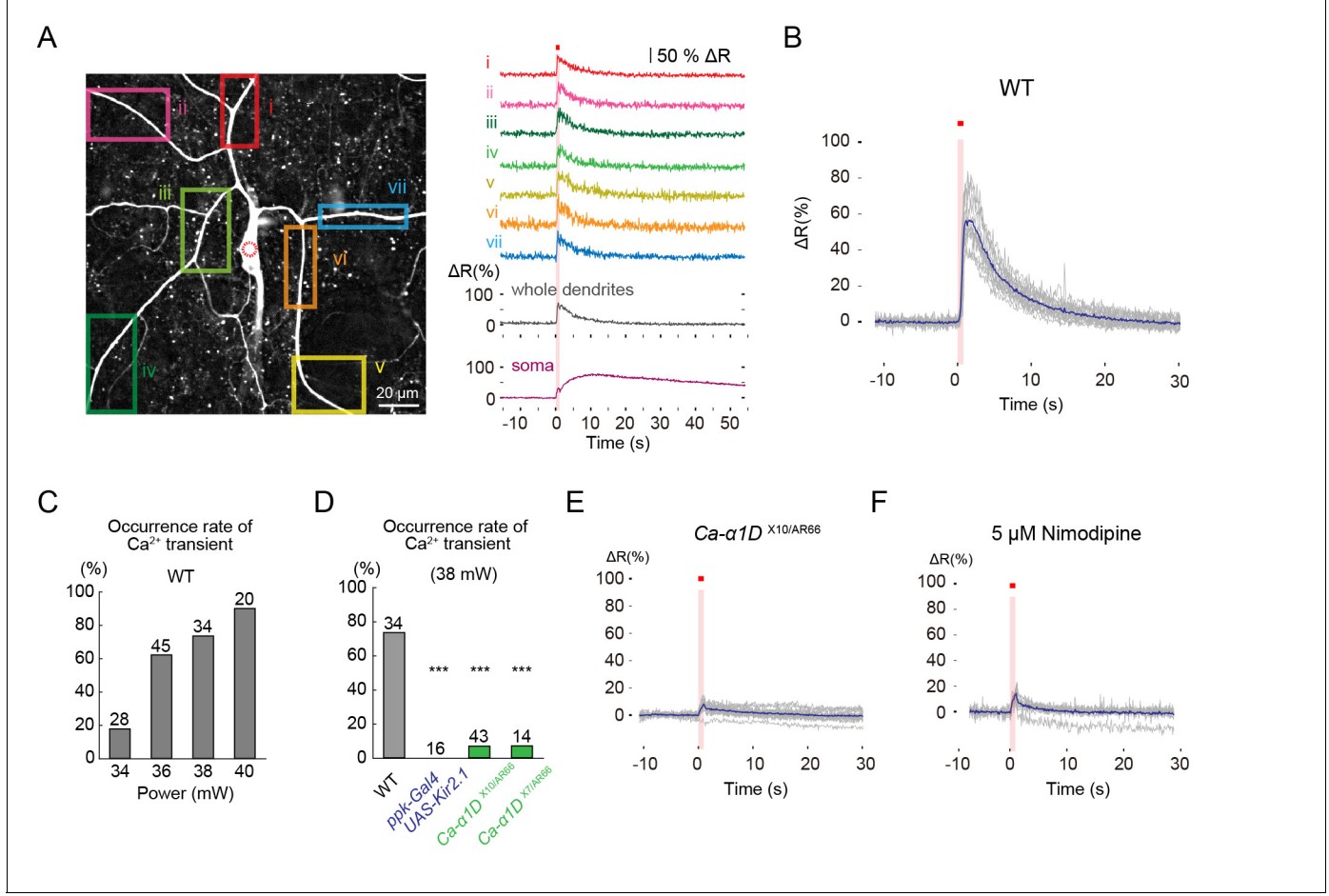

**Figure 2.** L-type VGCC-dependent global $Ca^{2+}$ transients occur in Class IV neurons upon noxious thermal stimulation. $Ca^{2+}$ responses of Class IV neuron ddaC expressing TN-XXL in whole-mount larvae, except for **F** (fillet preparations), upon IR irradiation. Throughout this figure, somata were targeted (red dotted circle in **A**). The output power of the laser was 38 mW except for **C**. Red squares above traces and magenta shadings in **A**, **B**, **E**, and **F** indicate the 1 s irradiations. (**A**) (Left) Time-projected image that is constructed by multiplying CFP and YFP images at every time point. Rectangles i–vii indicate the regions of interest (ROIs). (Right) Global $Ca^{2+}$ transients were detected in individual ROIs (i–vii). The transient was also detected when we drew a ROI around the entire dendritic arbor excluding the soma (whole dendrite). $Ca^{2+}$ transient measured in somata displayed slower fluctuation compared to the dendritic ones (soma). (**B**) Time courses of $Ca^{2+}$ transients in whole dendrites. Gray lines indicate dendritic $Ca^{2+}$ transients of 14 cells, and the blue line represents the averaged amplitude. The data from cells that did not generate $Ca^{2+}$ transients were excluded. (**C**) Occurrence rate of dendritic $Ca^{2+}$ transients when stimulated with different IR-laser output powers. We performed one trial per cell and the number of cells examined is displayed at the top of each bar. (**D**) Occurrence rate of dendritic $Ca^{2+}$ transients was dramatically reduced in Class IV neurons overexpressing Kir2.1 (*ppk-Gal4 UAS-Kir2.1*) or in neurons of larvae with mutations of the L-type VGCC gene (*Ca-α1D*[X10/AR66] and *Ca-α1D*[X7/AR66]). *** p < 0.001 versus wild type by Fisher's exact test. (**E**) $Ca^{2+}$ fluctuations in dendrites of *Ca-α1D* mutant neurons (*Ca-α1D*[X10/AR66]). Gray lines indicate $Ca^{2+}$ responses of 17 cells, and the blue line represents the averaged amplitude. (**F**) $Ca^{2+}$ fluctuations in dendrites of fillet preparations when treated with 5 μM Nimodipine. Gray lines indicate $Ca^{2+}$ responses of 11 cells, and the blue line represents the averaged amplitude. In **E** and **F**, we excluded data where ratiometric signals could not be continuously recorded due to movements of mounted larvae.

The following source data and figure supplements are available for figure 2:

**Source data 1.** A sample file of FRET imaging.

**Figure supplement 1.** Configuration of the region-of-interest (ROI) in $Ca^{2+}$ FRET imaging.

**Figure supplement 2.** $Ca^{2+}$ transients evoked by IR-laser irradiation of dendrites.

**Figure supplement 3.** Effects of genotypes and pharmacological treatments on occurrence rates of dendritic $Ca^{2+}$ transients, peak amplitudes of the transients, and/or slow $Ca^{2+}$ rises in somata, when somata were irradiated.

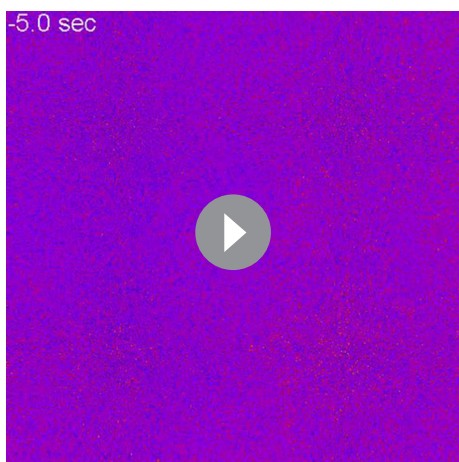

**Video 1.** Dendritic Ca$^{2+}$ transients (1). Related to *Figure 2*. Ratiometric pseudocolor time-lapse images of TN-XXL-expressing Class IV neuron ddaC in a whole-mount preparation of a larva. The time course is indicated in the upper left corner of each image, and the duration of IR irradiation (1 s at 38 mW) is displayed by red squares in upper right corner of relevant images. Genotype: *3×[ppk-TN-XXL] (attP2)/ 3×[ppk-TN-XXL] (attP2)*.

and distal dendrites, and decayed exponentially. Dendritic Ca$^{2+}$ transients were also visualized by using GCaMP5G (*Akerboom et al., 2012*) as an indicator (*Video 2*), but we used TN-XXL throughout this study for monitoring Ca$^{2+}$ dynamics in both whole-mount larvae and fillet preparations, which allowed more robust quantitative analysis in the presence of perturbations along Z-axis motions by body wall muscles.

The Ca$^{2+}$ transients occurred in all-or-none fashion upon IR-laser irradiations; suggesting that the magnitude of the thermal stimuli was converted into another value of information (e.g. dendritic membrane potential), and the value higher than a presumptive threshold evoked Ca$^{2+}$ transients. The probabilities of the occurrence of Ca$^{2+}$ transients were dependent on the output powers of the IR-laser, which arose abruptly at around 36 mW (*Figure 2C*), so this high occurrence is one advantage of irradiation to the soma. Furthermore, the probabilities were significantly lowered in the neurons of *dTrpA1* or *pain* mutants (*Figure 2—figure supplement 3A*), indicating that Ca$^{2+}$ transients also reflect normal neuronal activities that depend on the heat-activated channels. Irradiation to the dendritic arbor also evoked Ca$^{2+}$ transients, although this required higher IR-laser power compared to the stimulation of the soma (*Figure 2—figure supplement 2*). Once a Class IV neuron elicited a transient, that neuron then responded more weakly to the repetitive IR-laser irradiations onto the same location in the cell (data not shown). In contrast to the sharp Ca$^{2+}$ rises throughout dendrites upon IR irradiations to somata, Ca$^{2+}$ fluctuations within the somata were relatively slow (*Figure 2A*; 'soma' and *Figure 2—figure supplement 3B*).

## The L-type voltage-gated calcium channel is necessary for the dendritic Ca$^{2+}$ transient

We next explored the molecular basis of the dendritic Ca$^{2+}$ transient. Excitability of Class IV neurons was prerequisite for this Ca$^{2+}$ transient, as shown by the fact that the occurrence of the transients was completely blocked by overexpression of the inward-rectifier potassium channel Kir2.1 (*Figure 2D*), which hyperpolarizes neurons and decreases their resting membrane potential (*Hardie et al., 2001*). This raised the possibility that the transient was caused by Ca$^{2+}$ influx from extracellular space (see another possibility discussed in *Figure 2—figure supplement 3* and its legend) and that some of the voltage-gated Ca$^{2+}$ channels (VGCCs) were involved. To verify these possibilities, we screened genes encoding α subunits of VGCCs for those necessary for the occurrence of Ca$^{2+}$ transients; specifically, we recorded and analyzed probabilities and

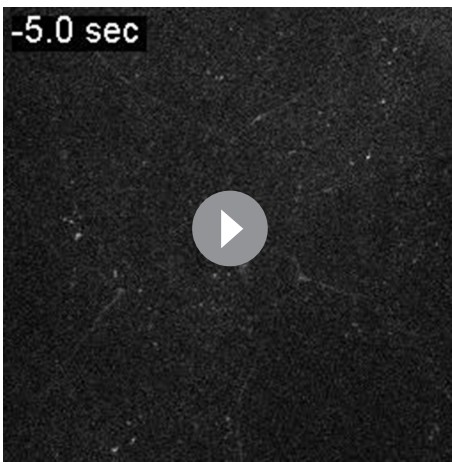

**Video 2.** Dendritic Ca$^{2+}$ transients (2). Related to *Figure 2*. Intensiometric unicolor time-lapse images of GCaMP5G-expressing Class IV neuron ddaC in a whole-mount preparation of a larva. The time course is indicated in the upper left corner of each image, and the duration of IR irradiation (1 s at 38 mW) is displayed by red squares in upper right corner of relevant images. Genotype: *UAS-GCaMP5G/+; ppk-Gal4/+*.

amplitudes of the $Ca^{2+}$ transients in various VGCC loss-of-function mutants. A dramatic phenotype was found in mutants of the L-type VGCC gene ($Ca$-$\alpha$$1D^{X7/AR66}$ and $Ca$-$\alpha$$1D^{X10/AR66}$)(*Eberl et al., 1998*); normal $Ca^{2+}$ transients hardly occurred in those mutants (*Figure 2D and E*). A similar defect was observed in the wild type when a selective inhibitor of L-type VGCC, Nimodipine (*Gielow et al., 1995*; *Grey and Burrell, 2010*), was applied (*Figure 2F*). These results showed that L-type VGCC-mediated excitability is required for the occurrence of $Ca^{2+}$ transients in Class IV neurons (see our negative results of other types of the *Drosophila* VGCC family in 'Materials and methods').

As for mammalian neurons, it has been well known that there is a link between changes in $Ca^{2+}$ levels, which are regulated by L-type VGCC, and cell-specific gene expression (*Greer and Green-berg, 2008*). Therefore, the expression level of *dTrpA1* and/or *pain* could have been lowered in the L-type VGCC mutants. To test this possibility, we performed real-time PCR to quantitate the mRNA levels of *dTrpA1* and *pain* in neural tissues of *Ca*-$\alpha$$1D$ mutants, but found no significant differences between the *Ca*-$\alpha$$1D$ mutant and the wild type (data not shown).

The $Ca^{2+}$ transient in Class IV neurons was reminiscent of that in rat neocortical pyramidal neurons, which are also detectable throughout the entire dendritic arbor (*Schiller et al., 1995*). The occurrence of this transient in the pyramidal neurons requires voltage-gated $Na^+$ channels (VGSCs) that are distributed along dendritic branches. In our system, however, the addition of Tetrodotoxin (TTX) had no effect on the dendritic $Ca^{2+}$ transients in Class IV neurons (*Figure 2—figure supplement 3C and E*), indicating that VGSCs are required neither for the occurrence of $Ca^{2+}$ transients nor for the regenerative propagation of membrane potential along the dendritic branches.

## Dendritic $Ca^{2+}$ transients are most likely generated by accumulations of $Ca^{2+}$ spikes producing the pauses in firing

To elucidate the relationship between $Ca^{2+}$ transients and neuronal activities, we performed extracellular single-unit recording and the ratiometric $Ca^{2+}$ imaging simultaneously (*Figure 3A*). Throughout our simultaneous recordings in this study, the IR stimulation was given to the proximal dendrite. Under 40-mW IR-laser irradiation, peak amplitudes of $Ca^{2+}$ transients did not display a strong correlation with maximum firing rates (*Figure 3B*; p > 0.46, $\rho$ = 0.17, Spearman's rank correlation test; *n* = 20 cells). In our analyses, the firing rates were computed by sliding a rectangular window function along the spike train with $\Delta t$ = 400 ms (see 'Materials' and methods for details). Even when we increased the width of the sliding window ($\Delta t$) from 10 to 1000 ms in steps of 10 ms, there were no statistically significant correlations between the estimated maximum firing rates and the peak amplitudes of $Ca^{2+}$ transients (p $\geq$ 0.1; *Figure 3—figure supplement 1A*).

Upon closer inspection of the firing patterns with $Ca^{2+}$ transients, we noticed characteristic pauses in firing that followed the high-frequency spikes (*Figure 3A*). Thus, we defined these spikes as a pair of unconventional spikes (US) based on the interspike interval (*Figure 3C*), and designated the firing pattern composed of the pause and US tentatively as a 'burst and pause' pattern. When we sorted datasets of the firing patterns in a descending order of the peak amplitude of $Ca^{2+}$ transients, the occurrence of US showed a high correlation with the peak amplitude (*Figure 3D and E*; p < 1.9$\times$10$^{-8}$, $\rho$ = 0.91; Spearman's rank correlation test; *n* = 20 cells). We also performed a *post hoc* parameter fitting analysis on the basis of our finding that the occurrence of the US showed a high correlation with the peak amplitude of $Ca^{2+}$ transients (*Figure 3E*; see also its legend). More importantly, in each dataset, the minimal interspike interval within the US was invariably shorter than that within non-US firings (*Figure 3F*; p < 5.2$\times$10$^{-6}$; paired Student's *t*-test; *n* = 13 cells), indicating that large excitatory currents existed just prior to the pause in firing (see also *Figure 3—figure supplement 1*).

To address how inhibition of L-type VGCC affects the firing rate and the occurrence of $Ca^{2+}$ transients and US, we recorded Class IV neurons in the presence of Nimodipine. $Ca^{2+}$ transients and US were hardly detected in the spike trains (*Figure 3G,H, and J*; [Control] 1.6 ± 0.32, [Nimodipine] 0 ± 0; p = 0.001, Wilcoxon's rank-sum test; *n* = 20 cells [Control] and *n* = 11 cells [Nimodipine]); in contrast, the firing rate was not decreased compared to that of the control and showed its peak around 100 Hz (*Figure 3I*). In other words, firing patterns were modified as if the 'burst and pause' pattern had been replaced with conventional high-frequency spike trains, indicating that the L-type VGCC is essential for the generation of US. We also found that firing rates decreased for short periods following the pauses (data not shown); therefore, we assumed that an after-hyperpolarization current is generated during the pauses (see 'Discussion'). Collectively, our findings strongly suggest that

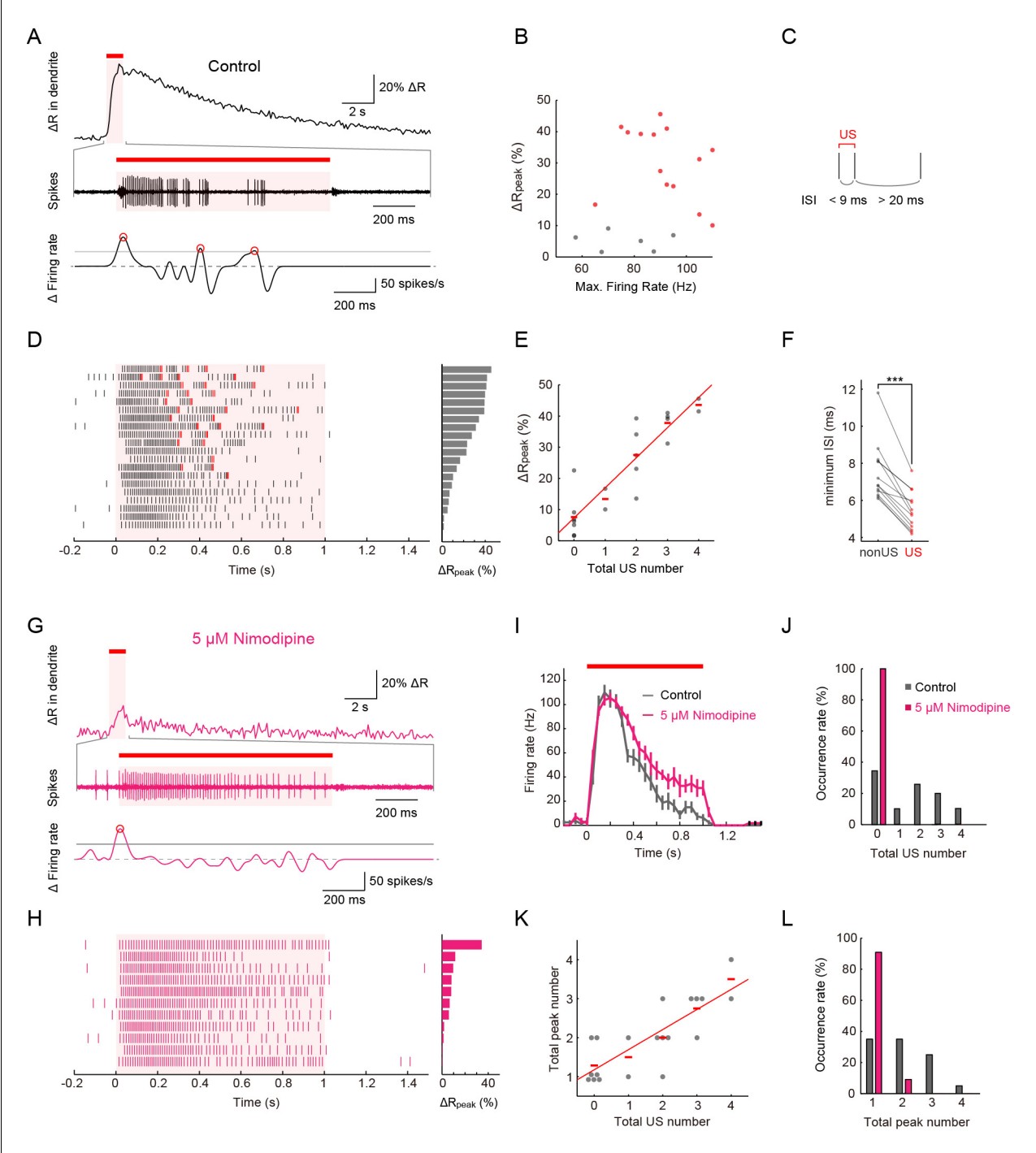

**Figure 3.** Simultaneous recordings of firing responses and dendritic Ca$^{2+}$ transients in fillet preparations of larvae upon noxious thermal stimulation. Data obtained from fillet preparations of control larvae (**A–F**), those from L-type VGCC-blocked larva (**G** and **H**), and their comparisons (**I–L**). In each trial, 40-mW IR laser was focused onto a proximal dendritic branch for 1 s (indicated by red bars above traces, and magenta shadings in (**A, D, G, I** and **H**) and the Ca$^{2+}$ transient was detected from a ROI that was set on a distal branch 100 μm away from the focus of the IR-laser beam. (**A** and **G**) Representative recordings of a control larva (**A**) and a larva treated with 5 μM Nimodipine (**G**). Dendritic Ca$^{2+}$ fluctuations for 20 s (top) and an enlarged trace of an extracellular recording for 1.675 s including the 1 s duration of the IR-laser irradiation (middle). The time derivative of firing rate fluctuation based on the spike density estimation with the Gaussian kernel (σ = 25 ms) and the normalized threshold settings (bottom). The peak of firing rate fluctuation is defined as in the text and marked with red circles. (**B**) Peak amplitudes of Ca$^{2+}$ transients are plotted against maximum firing rates (defined in Materials and methods). ΔR$_{peak}$ is defined as the averaged ratio of 400–700 ms after the cessation of IR-laser stimulation, which is employed as a representative value of each trial. See also the legend of **D**. A significant correlation was not observed (p > 0.46, ρ = 0.17; Spearman's rank

*Figure 3 continued on next page*

*Figure 3 continued*

correlation test; n = 20 cells). (C) The definition of 'unconventional spikes' (US). Signatures of US were extracted according to the following two conditions: (1) the first interspike interval (ISI) of three sequential spikes was less than 9 ms, and (2) the second ISI was longer than 20 ms. Then, a pair of the first and second spikes within each sorted triplet was designated as 'US'. (D and H) Raster plots of firing (left) and magnitudes of $\Delta R_{peak}$ (right). (D) All trials in panel (B), where control larvae were used, are sorted in a descending order of magnitude of $\Delta R_{peak}$ (right gray bars). Red raster lines indicate US. Six trials that elicited no US are labeled by gray color in B. (H) When fillet preparations were treated with 5 µM Nimodipine, none of the 11 trials elicited US during IR irradiation. (E) Amplitudes of $\Delta R_{peak}$ are plotted against total US numbers for each trial. Short horizontal red bars indicate the averages of $\Delta R_{peak}$, and the red line is a linear regression of plotted data (p < $1.9 \times 10^{-8}$, $\rho$ = 0.91; Spearman's rank correlation test; n = 20 cells). Our original definition of US in fact displayed a suboptimal correlation to the peak amplitude. We redefined USs using different parameter sets (lengths of the first and second interspike intervals) and calculated Spearman's correlation coefficients to estimate how well the total US number is correlated with the peak amplitude. As a result, our original definition in C in fact displayed a suboptimal correlation to the peak amplitude (data not shown). (F) Comparative analysis of ISI between US and between non-US events. For trials with at least one US, the minimum length of ISI between US in each trial was compared to that between non-US (p < $5.2 \times 10^{-6}$; paired t-test; n = 13 cells). (I–L) Temporal patterns of firing rates (I) and histograms of the total US number (J) or the total peak number (L) in control and Nimodipine-treated larva (gray and magenta, respectively). See also the legend of A. (K) A plot of total peak numbers versus total US numbers for each trial in the control. Data in I are presented as mean ± s.e.m.

The following figure supplements are available for figure 3:

**Figure supplement 1.** Simultaneous recordings of firing responses and dendritic $Ca^{2+}$ transients in fillet preparations of larvae upon noxious thermal stimulation.

**Figure supplement 2.** Simultaneous recordings of firing responses and dendritic $Ca^{2+}$ transients in fillet preparations of larvae expressing the lowest-threshold form of dTrpA1.

**Figure supplement 3.** Dual recordings of firing responses from the soma and the axon bundle.

dendritic $Ca^{2+}$ transients we observed in response to heat produced by the IR laser are generated by accumulations of L-type VGCC-dependent $Ca^{2+}$ spikes in Class IV neurons.

## Ectopic expression of the lowest-threshold form of dTrpA1 allows Class IV neurons to evoke both $Ca^{2+}$ transients and the 'burst and pause' firing pattern with lower powers of IR

The ectopic expression of thermoTRP has been widely used as a tool to achieve conditional controls of neuronal activity in vivo by modulating the ambient temperature (*Bernstein et al., 2012*). This is probably also the case with the ectopic expression of the lowest-threshold isoform of dTrpA1, one of the thermoTRPs described above, in Class IV neurons. *dTrpA1* encodes four distinct isoforms that display different temperature sensitivities (*Zhong et al., 2012*); among them, the dTrpA1-A isoform (*Viswanath et al., 2003*) (also designated as dTrpA1.K) (*Hamada et al., 2008*) has the lowest threshold (~29°C) and shows little desensitization by repetitive heat-activations (*Pulver et al., 2009*). Forced expression of dTrpA1-A in Class IV neurons allows larvae to respond to a 30°C heat probe and elicit the nocifensive rolling behavior (*Babcock et al., 2011*; *Zhong et al., 2012*), possibly by endowing the neurons with higher sensitivity to the moderate-temperature stimulus. Consistent with this speculation, our recordings demonstrated that TrpA1-A-expressing neurons (TrpA1-A[+] neurons) evoke both $Ca^{2+}$ transients and the unique firing pattern below 20 mW (*Figure 3—figure supplement 2A*), to which the wild-type neurons were insensitive (*Figure 1D*), and all of other results (*Figure 3—figure supplement 2B–E*) strengthened the possibility that the cellular responses of Class IV neurons to IR irradiation reflected the intrinsic membrane properties.

## Spikes that are detected by extracellular single-unit recordings of soma propagate faithfully through its axon

Are the spikes detected in somata by extracellular single-unit recordings faithfully propagated through their axons? This had to be verified because of concerns in previous studies. For example, when neuronal activities with $Ca^{2+}$ spikes are recorded extracellularly, the intracellularly detected spikes sometimes disappear in the voltage traces of extracellular single-unit recordings of axons (*Schonewille et al., 2006*). Another instance shows that intracellularly detected spikes are not always propagated along the axon (*Monsivais et al., 2005*). To explore these possibilities, we performed

extracellular single-unit recordings from both the soma of a Class IV neuron and the axon bundle, including its own axon, and analyzed the correspondence between the firing patterns from somata, which coincided with US, and those from axon bundles (*Figure 3—figure supplement 3A*). Spikes from a single axon bundle can include those of not only Class IV but also other sensory neurons. Nonetheless, our findings indicate that the spikes recorded extracellularly from Class IV soma propagate faithfully through its axon (*Figure 3—figure supplement 3B–D*, see also the legends).

## The *Ca-α1D* gene expression in Class IV neurons is necessary for the robust induction of larval nocifensive rolling behavior

We hypothesized that the unique firing patterns accompanied by $Ca^{2+}$ influx should be output signals provoking the robust nocifensive rolling behavior. To verify this hypothesis, we conducted a series of physiological and behavioral assays of *Ca-α1D* mutant or knockdown larvae (*Figure 4* and *Figure 4—figure supplement 1*). We first examined the physiological responses of *Ca-α1D*-knockdown Class IV neurons. Both the occurrence of the $Ca^{2+}$ transient and that of the unique firing pattern were suppressed in the neurons (*Figure 4A and B*) without a dramatic decrease in firing rate (*Figure 4—figure supplement 1*), as was seen in the case of pharmacological blockade of L-type VGCC (*Figure 3G–I*). We also observed that the total US number was significantly decreased in *Ca-α1D*-knockdown neurons compared to controls (*Figure 4C*; [*white RNAi* [GL00094]] $1.55 \pm 0.20$ total US number, [*Ca-α1D RNAi* [HMS00294]] $1.00 \pm 0.00$; p = 0.0055; Wilcoxon's rank-sum test).

We then examined how *Ca-α1D* mutant larvae (*Ca-α1D*[X10/AR66]) responded to the heat probe and found that they displayed a significantly delayed response (*Figure 4E*), almost identical to *dTrpA1* mutants (*Babcock et al., 2011*; *Neely et al., 2011*) (our data not shown). However it was reported that synaptic transmission at the neuromuscular junction (NMJ) is impaired in the *Ca-α1D* mutant larvae due to abrogation of the $Ca^{2+}$ influx in both muscles and motor neurons (*Ren et al., 1998*; *Worrell and Levine, 2008*), so it was difficult to attribute the behavioral defect entirely to the role of L-type VGCC in Class IV neurons. Therefore, we knocked down expression of L-type VGCC selectively in Class IV neurons as was previously done (*Kanamori et al., 2013*). The phenotypic defect of the knockdown larvae was less dramatic than that of the whole-body mutant; nonetheless, the knockdown larvae displayed significantly delayed response in the rolling behavior than the control larvae (*Figure 4F*). These data support our hypothesis that L-type VGCC-dependent unique firing patterns in Class IV neurons are necessary for the highly penetrant induction of the larval nocifensive rolling behavior.

## Optogenetic experiments strongly suggest that multiple fluctuations in firing rate in Class IV neurons enhances robust rolling behavior

To explore the possibility of causal connections between the 'burst and pause' firing pattern and the rolling behavior, we performed a series of optogenetic activation experiments (*Figure 5*). It has been shown that larvae expressing channelrhodopsin-2 (ChR2) in Class IV neurons mimic the rolling behavior when ChR2 is activated by continuous light exposure (*Hwang et al., 2007*). We have established our assay of the induced rolling by expressing a faster kinetic variant of ChR2, ChIEF in Class IV neurons (*Mattis et al., 2012*). To address whether ChIEF-dependent intermittent firing of Class IV neurons is sufficient to elicit the rolling behavior, whereas continuous firing is not, we tried to recapitulate the naturalistic 'burst and pause' firings in the background of a Class IV-specific *Ca-α1D*-knockdown by activating ChIEF using square pulses of light. On and off durations of light pulses were set to be comparable to 'burst' and 'pause' timings (100 ms and 100 ms, respectively; *Figure 5A* top).

With these illumination parameters, either continuous or intermittent firing patterns were successfully evoked in the knockdown neurons of fillet preparations (top of *Figure 5A*). In live animals, *Ca-α1D*-knockdown larvae displayed rolling when the continuous firing pattern was evoked, and the rolling occurred more efficiently by the pulsatile optogenetic activation (*Figure 5B*; [Continuous] $65.3 \pm 9.2\%$, n = 75, [Intermittent] $86.3 \pm 5.9\%$, n = 95, mean $\pm$ 95% confidence interval based on Clopper-Pearson method; p = 0.0017, Fisher's exact test). Although the probability of rolling was rather high in the continuous activation condition (see 'Discussion'), this result is, nevertheless, consistent with the idea that the intermittent firing pattern facilitates the rolling behavior. However, it should be stressed that the intermittent pattern produced under our optogenetic experiment was distinct from

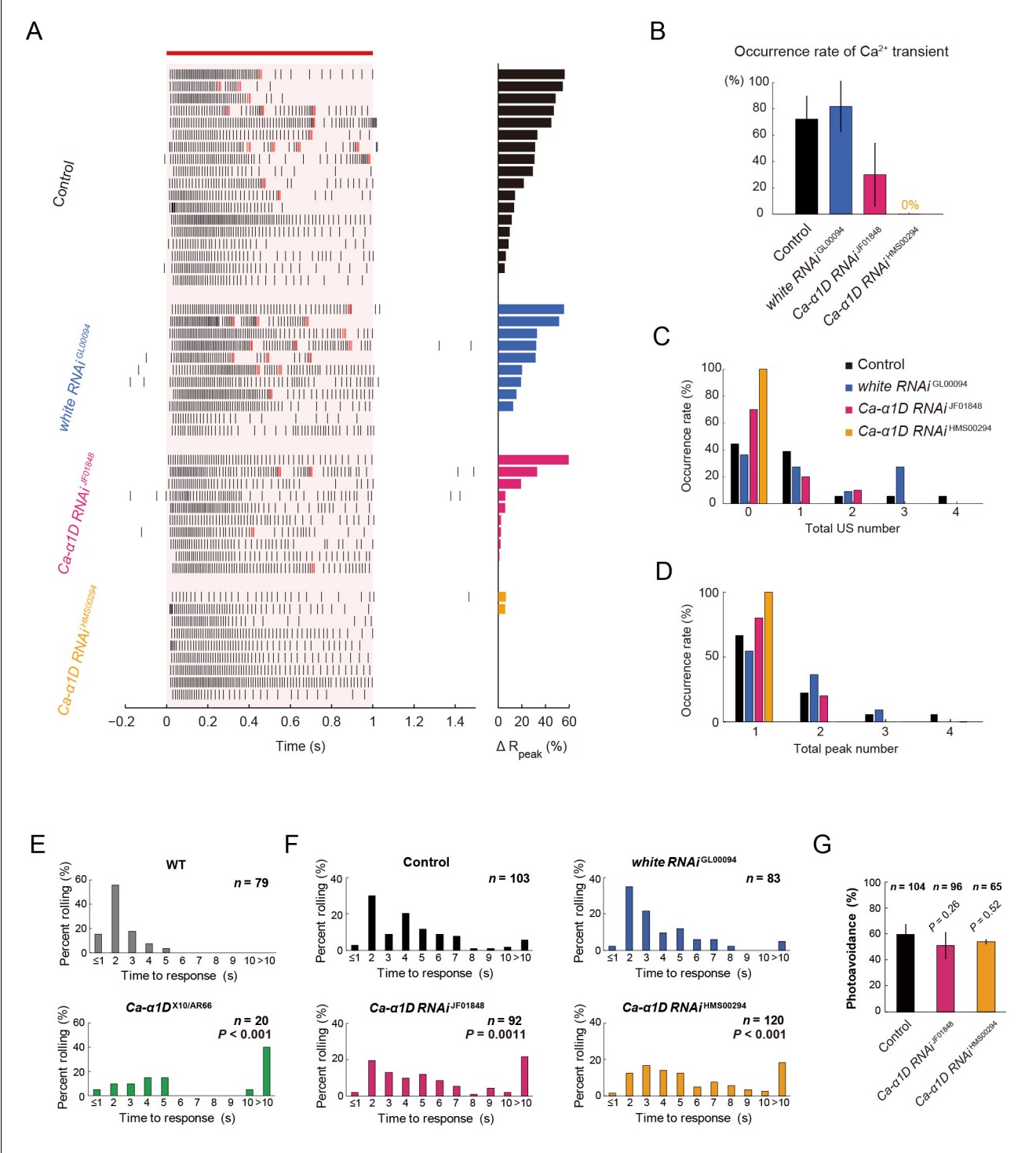

**Figure 4.** Responses of Class IV neurons with a knockdown of *Ca-α1D* and behavioral analysis of the knockdown larvae upon noxious thermal stimulation. (A–D) Responses of control Class IV neurons (Control, *n* = 18 and *white RNAi* [GL00094], *n* = 11) and *Ca-α1D* knockdown neurons (*Ca-α1D RNAi* [JF01848], *n* = 10 and *Ca-α1D RNAi* [HMS00294], *n* = 9) to 1 s irradiation of a 48-mW IR laser. (A) Raster plots of firing (left) and magnitudes of $\Delta R_{peak}$ corresponding to Ca$^{2+}$ transients (right). Trials are sorted in a descending order of the magnitude of $\Delta R_{peak}$ (right bars). Red bar indicates the 1 s irradiation and red raster lines are US. (B) Occurrence rate of dendritic Ca$^{2+}$ transients in each genotype([Control] 72.2 ± 17.6%, [white RNAi GL00094] 81.8 ± 19.4%, [*Ca-α1D RNAi* [JF01848]] 30.0 ± 24.2%, [*Ca-α1D RNAi* [HMS00294]] 0.0 ± 0.0%; mean ± 95% confidence interval [Clopper-Pearson method]). (C and D) Histograms of the total US number (C) or the total peak number (D) during the irradiation in the control and the knockdown neurons. The total peak number of firing rate fluctuation is defined as in the text. See also *Figure 3A* and its legend. (E and F) The distribution of nocifensive escape

*Figure 4 continued on next page*

*Figure 4 continued*

locomotion latency for wandering third-instar larvae of the wild type and the *Ca-α1D* mutant that were stimulated with a 47°C probe. The numbers of larvae tested are indicated and data are presented as percentage. (E) The distributions of the wild-type and *Ca-α1D* mutant larvae (mean latency: [w[1118]] 1.78 s, [*Ca-α1D* [AR66/X10]] 6.12 s; Wilcoxon rank-sum test). (F) The distributions of control larvae and larvae with Class IV neuron-specific *white* or *Ca-α1D* knockdown. The *Ca-α1D* mutant and the *Ca-α1D* knockdown larvae showed significant delayed nocifensive responses in comparison to the wild type (E) and the controls (Control and *white RNAi* [GL00094]) (F), respectively. p values in F are those of the test between *white RNAi* [GL00094] and each of the *Ca-α1D* knockdowns (mean latency: [Control] 3.87 s, [*white RNAi* [GL00094]] 3.39 s, [*Ca-α1D RNAi* [JF01848]] 5.21 s, [*Ca-α1D RNAi* [HMS00294]] 5.25 s; Wilcoxon rank-sum test with Bonferroni correction). (G) Percentage of larvae avoiding blue light (457–487 nm; 0.72 mW/mm$^2$; [Control] 59.6 ± 8.0%, [*Ca-α1D RNAi* [JF01848]] 53.9 ± 10.3%, [*Ca-α1D RNAi* [HMS00294]] 51.0 ± 8.5%; mean ± 95% confidence interval [Clopper-Pearson method]). We employed GMR-hid to ablate Bolwig's organs in the photo-avoidance assay. p values are indicated (two-tailed Fisher's exact test). Sixty-five to 104 larvae were tested for each condition.

The following source data and figure supplement are available for figure 4:

**Source data 1.** The distributions of nocifensive escape locomotion latency for wandering third-instar larvae of the wild-type and *Ca-α1D* mutant larvae (Columns A, B) and The distributions of control larvae and larvae with Class IV neuron-specific *white* or *Ca-α1D* knockdown (Column D–G).

**Figure supplement 1.** Simultaneous recordings of firing responses and dendritic Ca$^{2+}$ transients in fillet preparations of *Ca-α1D* knockdown larvae upon noxious thermal stimulation.

the pattern of the wild-type neuron upon noxious thermal stimulations according to two quantitative features. First, the maximum firing rate of the optogenetic pattern was 40~50 Hz (data not shown) and lower than that of wild-type neurons (~80 Hz; *Figure 1E*). Second, none of the spike trains generated in our optogenetic condition fully met the definition of US (*Figure 3C*). Therefore, we reconsidered our idea that the occurrence of the US is a reliable metric for the artificial firing pattern, and we searched for other metrics that would explain all of the results of our three experimental settings:

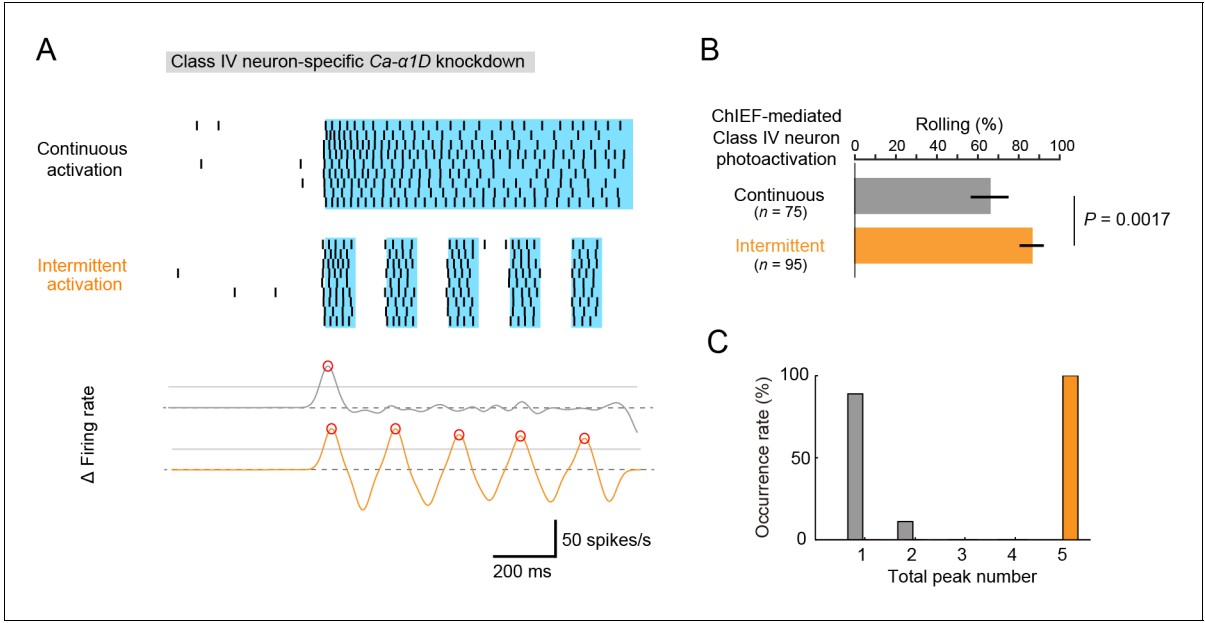

**Figure 5.** Optogenetic activation of Class IV neurons. (A) Data obtained from fillet preparations of larvae where *Ca-α1D* gene was knocked down specifically in Class IV neurons. (Top) Raster plots of firing. A faster kinetic variant of ChR2, ChIEF, was expressed selectively in the neurons and activated with a blue LED continuously ('Continuous activation'; *n* = 9) or in a pulsatile fashion ('Intermittent activation'; *n* = 9). (Bottom) The time derivative of firing rates under the continuous activation (gray) or intermittent activation (orange). The peak of firing rate fluctuation is defined as in the text and marked with red circles. (B) Percentage of larvae rolling their bodies upon optogenetic activation ([Continuous] 65.9 ± 11.5%, *n* = 75; [Intermittent] 88.5 ± 7.6%, *n* = 95; mean ± 95% confidence interval [Clopper-Pearson method]). p-Value is indicated (two-tailed Fisher's exact test). (C) A histogram of the total peak number during the continuous activation (gray) or intermittent activation (orange).

the effect of Nimodipine application (*Figure 3D and H*), the *Ca-α1D*-knockdown phenotypes (*Figure 4A*), and the optogenetics (the top of *Figure 5A*).

We noticed that fluctuations of firing rate were prominent around the time when each US occurred (the recording trace of Δ Firing rate of *Figure 3A*; see also *Figure 3D*). To express such fluctuations quantitatively, we first speculated that the number of the peaks of the firing rate during IR stimuli might be an appropriate metric and computed the firing rate by using the Gaussian kernel method (*Figure 3—figure supplement 1C*; see also below). Although we defined the peak of the firing rate using different parameter sets, we did not find statistically significant differences in the peak number between the controls and the samples in the above three experimental settings (data not shown).

Next, we focused our attention on the local steepness of the firing rate fluctuation during heat stimulation because a precipitous change in the firing rate is one essential component of burst firing. To quantify 'up' and 'down' of the firing rate fluctuation, we calculated the time derivative of the firing rate fluctuation during IR irradiation, defined peaks of firing rate fluctuation, and scored the total number of the peaks ('total peak number') as follows: first, the firing rate estimation was computed by using the Gaussian kernel method ($\sigma$ = 25 ms); second, a threshold was configured as 0.5 of the primal peak value; finally, peaks of the time derivative above the threshold were specified (red circles in the bottom graphs of *Figure 3A,G* and *5A*). With this analysis, the 'total peak number' showed a significant correlation with the total US number in control neurons (p = 3.17 x $10^{-5}$, $\rho$ = 0.79; Spearman's rank correlation test; *n* = 20 cells; *Figure 3K*). We then showed a significant decrease in the total peak number in Nimodipine-applied neurons (*Figure 3L*; [Control] 2.00 ± 0.21 total peak number, [Nimodipine] 1.09 ± 0.091; p = 0.0034; Wilcoxon's rank-sum test). Furthermore, the total peak number was significantly decreased in *Ca-α1D*-knockdown neurons compared to controls (*Figure 4D*; [white RNAi $^{GL00094}$] 1.55 ± 0.20 total peak number, [Ca-α1D RNAi $^{HMS00294}$] 1.00 ± 0.00; p = 0.027). As is intuitively obvious in the optogenetics experiment, the total peak number under the continuous activation was significantly different from that of the intermittent one (*Figure 5C*; [Continuous] 1.11 ± 0.11, [Intermittent] 5.00 ± 0.00; p = 4.11 x $10^{-5}$).

To conclude, we consider the 'total peak number' is a reliable metric that can represent the complex burst and pause patterns and designate the peak of the time derivative of firing rate fluctuation as the peak of firing rate fluctuation for short. These results suggest that the total number of peaks of firing rate fluctuation in Class IV neurons plays a key role in increasing the likelihood of rolling locomotion upon noxious heat input. We designate this hypothesis as the 'burst number-coding' model because the burst (i.e., high-frequency firing) is accompanied by the peak of firing rate fluctuation.

To explore another potential impact of the 'burst and pause' firing pattern on escape behaviors, we tested whether it plays an important role in evoking the 'fast crawling' after noxious heat stimulation (*Ohyama et al. 2013*; *2015*). To evaluate a causal link between the firing patterns and the fast crawling, we performed a set of optogenetic experiments as follows: First, we confirmed that ChIEF-expressing Class IV neurons could be activated with lower-power blue light (0.056 mW/mm²; 470 nm) and displayed lower-frequency firings (10~20 Hz) with no 'burst and pause' pattern in *Ca-α1D*-knockdown larvae (data not shown). Then, we observed that such a firing pattern triggered the fast crawling, even when no rolling behavior was evoked (Maximum speed of crawling stride: [before activation] 6.37 ± 0.29 mm/s, [after activation] 10.51 ± 0.66 mm/s, mean ± s.e.m., *n* = 24 larvae; p = 0.0269, Wilcoxon signed-rank test; see 'Materials and methods' for details). These findings indicate that lower frequency firing is sufficient for eliciting the fast crawling, whereas the occurrence of 'burst and pause' firing pattern is not required.

## The *Ca-α1D* gene expression in Class IV neurons is not required for photo-avoidance behavior

Finally we addressed two questions that are relevant to the polymodality of Class IV neurons (*Xiang et al., 2010*). The first one was whether the L-type VGCC is also required for the avoidance of the short-wavelength light stimuli (the directional shift of locomotion), for which dTrpA1-dependent neural activity of Class IV neurons was essential. *Ca-α1D*-knockdown larvae avoided blue light, as did the control larvae, when they were illuminated (*Figure 4G*). This finding indicates that not only L-type VGCC is dispensable for the photo-avoidance behavior, but also the synaptic transmission at the NMJ is not severely affected under our *Ca-α1D* knockdown condition.

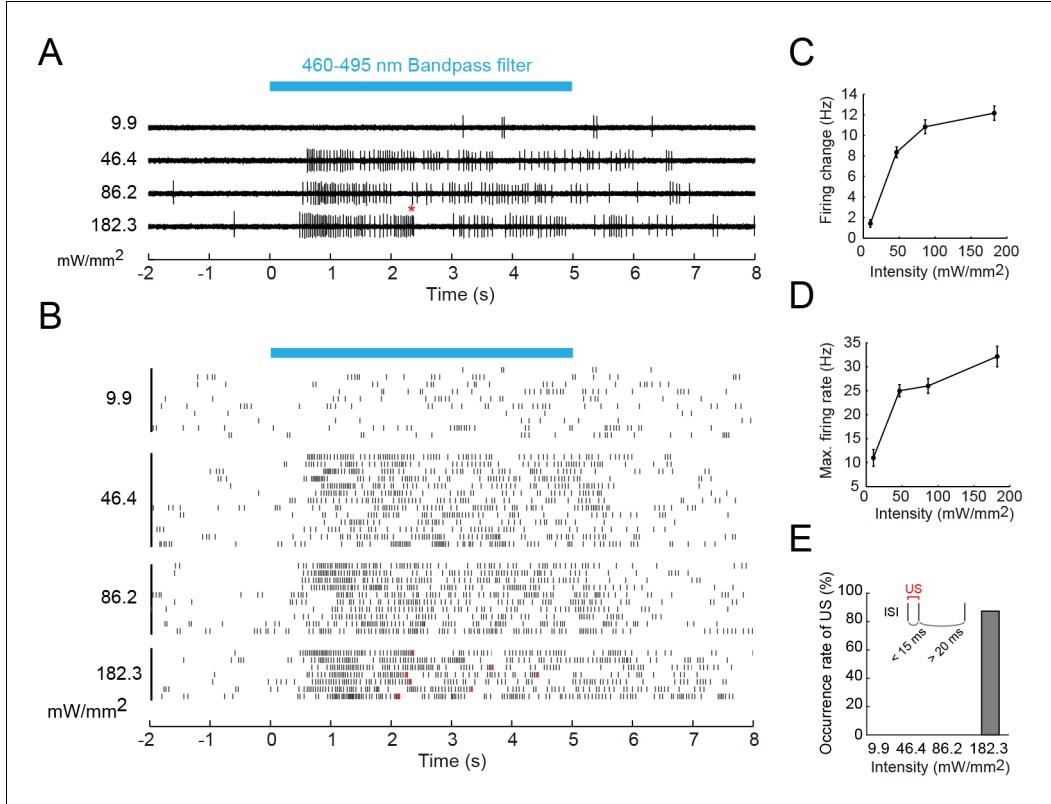

**Figure 6.** Firing responses of Class IV neurons upon illumination with blue light. Extracellular single-unit recordings of blue light (460–495 nm)-stimulated Class IV neurons in fillet-mounted larvae. (**A**) The blue bar above the traces indicates the 5 s duration of blue light illumination. Power densities of illumination are indicated on the left sides of individual traces (9.9–182.3 mW/mm$^2$). The red asterisk indicates a redefined unconventional spike (see US in the inset of **E** and the definition below). (**B**) Raster plots of firing with various power densities of illumination ($n$ = 7~13). Red tick marks in the responses to 182.3 mW/mm$^2$ indicate redefined US (see also **D**). (**C**) Firing frequency changes (average frequency of 5 s before blue light exposure subtracted from average frequency during 5 s of blue light exposure [*Xiang et al., 2010*]) are plotted against power density. (**D**) Maximum firing rates are plotted against intensity of blue light. Data are presented as mean ± s.e.m. (**E**) Occurrences of redefined US upon blue light illumination. US is redefined as follows: (1) the first interspike interval (ISI) of three sequential spikes is less than 15 ms, and (2) the second ISI is longer than 20 ms. Then, a pair of the first and second spikes within each sorted triplet is designated as 'redefined US' (inset).

The following figure supplement is available for figure 6:

**Figure supplement 1.** A model of the information processing: The polymodal encoding of thermo- and photo-nociception and decoding.

---

The second question was whether the wild-type Class IV evokes the 'burst and pause' firing pattern in response to the blue light as it did upon the noxious thermal stimuli. We illuminated larvae expressing TN-XXL in Class IV with short-wavelength light of four different intensities, and found the following (*Figure 6*): (1) The firing change and the Max firing rate increased with the light intensity, but the firing rate was 32.1 Hz even at the strongest intensity (182.3 mW/mm$^2$; *Figure 6A–D*), which exceeds the level that larvae normally experience in natural environments. This firing rate was far smaller than that in response to the thermal stimuli (83.9 ± 7.3 Hz; *Figure 1E*). (2) 'US-like spikes' (compare its definition in the inset of *Figure 6E* with that of US in *Figure 3C*) were detected only under the strongest light condition (*Figure 6E*). (3) Ca$^{2+}$ transients were not observed at a light intensity (146.7 mW/mm$^2$; data not shown). These observations indicate that the wild-type Class IV did not generate L-type VGCC-dependent US in response to the blue light.

Our results, together with previous studies, indicate that molecular mechanisms in Class IV under-lying avoidance to two distinct stimuli only partially overlap. Furthermore we propose the 'burst number-coding' hypothesis that, in Class IV neurons, the number of burst firings is employed as the facilitating signal for the heat stimuli and they instruct target neuron(s) to execute one out of the two possible behavioral outputs: the rolling behavior rather than the photo-avoidance (*Figure 6—figure supplement 1*; see 'Discussion').

## Discussion

### The presence of thermotransduction machinery in the dendrite

It is accepted that the phototransduction machinery exists in the dendrite of Class IV neurons (*Xiang et al., 2010*). However, previous protocols for delivering noxious thermal or mechanical stim-uli did not allow us to address whether the dendrite possesses transduction machineries for those stimuli. Our IR laser irradiation-measurement device is a noninvasive system, which is applicable to whole-mount animals and able to induce a rapid and local rise in the ambient temperature, mimick-ing the noxious heat input. With the resolution of our experimental conditions, the IR irradiation had no apparent effect on the integrity of organelles and evoked physiological responses of Class IV neu-rons. With the help of this system, we showed that the thermotransduction machinery is present in the dendrite.

### Identified and unidentified ion channels coordinate to generate the putative thermocurrent and the dendritic $Ca^{2+}$ transient

The hypothetical thermotransduction machinery includes dTrpA1 and Painless of the TRPA channel family, which are required for the stereotype avoidance behavior to noxious heat stimuli (*Neely et al., 2011*; *Tracey et al., 2003*). Class IV neurons in the *dTrpA1* mutant displayed a much longer latency time prior to the increase in the firing rate compared to the wild type. On the other hand, the rising profile of the firing rate in the *pain* mutant was comparable to that of the wild type, but the rate reached only 65% of that of the wild-type neurons. Therefore the temporal pattern of the wild-type firing response was not a simple summation of those of the two TRPA mutants. If dTrpA1 and Painless are the unique cation channels for the conductance of the putative thermocur-rent, the time course of the firing in *painless* mutants may represent that of dTrpA1 activation, and vice versa. If this inference is the case, our results can be interpreted in a way that dTrpA1 can become activated to some extent by itself and that it is prerequisite for Painless to be properly acti-vated. A previous electrophysiological analysis of Painless showed that Painless is a noxious-heat activated, $Ca^{2+}$-requiring, $Ca^{2+}$-permeable channel (*Sokabe et al., 2008*). Therefore, we suppose that noxious heat stimulation activates dTrpA1 and causes local $Ca^{2+}$ entry in the dendrite, followed by the robust increase in $Ca^{2+}$ conductance of Painless, leading to large depolarization. This results in gating of L-type VGCC and generation of the $Ca^{2+}$ transient, most likely by accumulated $Ca^{2+}$ spikes, throughout the dendrites of Class IV neurons.

Although little is known about subcellular localization of dTrpA1 and Painless in Class IV neurons (*Hwang et al., 2012*), they may distribute along the entire dendritic arbors. Among the subunits pre-sumably composing the functional VGCC in *Drosophila*, Straightjacket (Stj) of the α2δ family is expressed in da neurons and required for the heat avoidance (*Neely et al., 2010*). Thus, Stj may coexist with Ca-α1D in the dendrite. It remains to be elucidated how the burst is followed by the pause. We assumed that any of the $Ca^{2+}$-activated $K^+$ channels ($K_{Ca}$ channels) and/or other channels might be engaged in generating the pause through production of an after-hyperpolarization current.

### Roles of $Ca^{2+}$ spikes in nociception of Class IV neurons

In general, specialized sense organs such as nociceptors have thresholds at or near noxious levels, and increase activity with stronger noxious stimuli (*Perl, 2007*). Likewise, the probability of the $Ca^{2+}$ transient occurrence in Class IV neurons arose abruptly at around an IR-power that was estimated to raise the temperature close to 47°C, and the neuron increased the magnitude of the responses at the suprathreshold temperature. What would be the potential advantages for Class IV to set the threshold by employing $Ca^{2+}$ spikes? One advantage would be the improvement of the signal-to-noise ratio of the thermosensation. The cellular threshold of the response to the noxious heat could

become precipitous by multiplying a physical property of the high-threshold channel (dTrpA1) by the probability of the $Ca^{2+}$ spike generation.

Another advantage would be to utilize the generation of $Ca^{2+}$ spikes to winnow a heterogeneous modality in a selective manner. Our observations highlighted two features of the neuronal response to the different noxious stimuli: (1) the presence or the absence of multiple peaks of firing rate fluctuation and (2) the high or low absolute value of maximum firing rate. We hypothesize that the $Ca^{2+}$ spike can be a key signal encoding a specific modality: the total number of peaks of firing rate fluctuation transmits the noxious heat input by way of $Ca^{2+}$ spikes; in contrast, the continuous firing train at a lower rate encodes the noxious light sensation (e.g. see *Figure 1E* and *6D*). This means that the thermal stimuli induce the high-threshold and larger generator currents (orange line; *Figure 6—figure supplement 1* bottom), while blue light inputs evoke the low-threshold and smaller currents (blue line; *Figure 6—figure supplement 1* bottom). We showed that the heat-induced unique firing patterns of Class IV were conducted along the axon, which projects to the putative central pattern generator(s) driving the thermal nocifensive responses. It has been shown that Channelrhodopsin2-mediated activation of Class IV provokes the thermal nocifensive behavior (*Hwang et al., 2007*) and the photo-avoidance (*Xiang et al., 2010*). We predict that inward currents would be larger in the former.

Our optogenetic activation experiment shows that the multiple peaks of firing rate fluctuation is not essential for the nocifensive escape locomotion (rolling), but enhances its occurrence rate. Therefore, we speculate that such unique firing rate fluctuation may be decoded as a facilitating signal of rolling (the 'burst number-coding' hypothesis). It has also been shown in olfactory behaviors of *C. elegans* that specific activity patterns may modulate the probability of action selection (*Gordus et al., 2015*). Recently, it has been reported that combining mechanosensory and nociceptive cues synergistically enhances the selection of rolling behavior in *Drosophila* larvae (*Ohyama et al., 2015*). Altogether, the likelihood of rolling locomotion upon noxious heat stimulation can be potentiated by the input(s) from other sensory modalities and multiple peaks of firing rate fluctuation. We also noticed that the probability of rolling was rather high in the continuous activation condition (66.0%; gray bar in *Figure 5B*). We assume that almost all of the Class IV neurons were activated under our optogenetic conditions, so that such a high behavioral reactivity was elicited. On the other hand, lower probabilities of rolling were detected in the initial periods of the heat probe assay of *Ca-α1D* knockdown larvae (~20%, <2 s; *Ca-α1D RNAi* in *Figure 4F*). This low reactivity can be explained by local heat-stimulation that may activate a small number of neurons (*Robertson et al., 2013*).

## How are the firing patterns discriminated downstream of Class IV neurons?

The neural circuits in the central nervous system must read out the activity of Class IV neurons and distinguish one firing pattern from another to decode the polymodal sensory input. How are the two types of firing patterns in response to the distinct noxious inputs decoded and transformed into differential escape behaviors? Previous studies have shown that the burst coding is a behaviorally relevant mechanism (*Krahe and Gabbiani, 2004*). For instance, the bursts in an ultrasound-sensitive acoustic sensory neuron of the field cricket *Teleogryllus oceanicus* code for a conspicuous increase in amplitude of an ultrasound stimulus (*Sabourin and Pollack, 2009*). It is well known that in the cerebellar cortex, climbing fiber inputs evoke characteristic complex spikes in Purkinje neurons, which induce the dendritic $Ca^{2+}$ entry through $Ca^{2+}$ spikes and a subsequent pause in the spike train (*Davie et al., 2008*; *Kitamura and Häusser, 2011*; *Llinas and Sugimori, 1980*; *Mathews et al., 2012*). The firing response of Purkinje neurons is remarkably similar to that of Class IV neurons in terms of the pattern and the underlying mechanism. It should be noted that the pause of firing in Purkinje neurons evokes the high-frequency firing response in the target neurons of the deep cerebellar nuclei via the post-inhibitory rebound (PIR) mechanism (*De Zeeuw et al., 2011*). Moreover, it has been proposed that the pauses of firings per se convey signals (the 'pause coding' theory, *De Schutter and Steuber, 2009*; *Hong and Optican, 2008*). Another circuit evoking the nocifensive light-avoidance behavior may receive direct excitatory inputs from Class IV neurons; this would be consistent with our hypothesis that the dual codes of the polymodal sensory information are decoded and segregated as distinctive command signals in the downstream circuit. Recently, the candidate synaptic partner(s) of Class IV neurons were identified through functional and anatomical

assays (*Vogelstein et al., 2014*; *Ohyama et al., 2015*); thus, our hypothesis can be tested by investigating the neuronal responses of these candidate neurons to noxious thermal stimulations of Class IV neurons.

## Materials and methods

### *Drosophila* mutant and transgenic strains

The following fly strains were generated and/or used: (1) The overall structure of our construct for expressing the FRET-based $Ca^{2+}$ indicator TN-XXL (*Mank et al., 2008*) in Class IV neurons was as follows: The TN-XXL ORF was flanked by 1.9 kb *pickpocket* (*ppk*) promoter (*Grueber et al., 2003*) and a SV40 polyA signal sequence. Three copies of this cassette with intervening gypsy insulators (*Ni et al., 2009*) were subcloned into a pCasper4-derived plasmid that contains a PhiC 31 *attB* site (*Groth et al., 2004*). The resulting construct (*3×[ppk-TN-XXL]*) was integrated into either *attP40* or *attP2*. (2) Mutants of thermoTRP genes were *pain$^1$* and *pain$^3$* (*Sakai et al., 2009*; *Tracey et al., 2003*) from T Sakai, and *dTrpA1$^1$*(*Kwon et al., 2008*) and *dTrpA1$^{ins}$*(*Hamada et al., 2008*) from P. Garrity. (3) Mutants of the L-type VGCC α subunit gene were *Ca-α1D$^{AR66}$* from C Duch, and *Ca-α 1D$^{X10}$* and *Ca-α1D$^{X7}$* (*Eberl et al., 1998*) from Bloomington Stock Center. (4) A mutant of the T-type VGCC α subunit gene was *Ca-α1T$^{del}$* (*Ryglewski et al., 2012*) from Bloomington Stock Center. (5) Mutants of the P/Q-type VGCC α subunit gene were *cacophony$^S$* (*cac$^S$*[*Smith et al., 1998*]) from RW Ordway and *cac$^{ts4}$*(*Rieckhof et al., 2003*) from JT Littleton. We examined the peak amplitude of thermo-dependent $Ca^{2+}$ transients in the above T-type and P/Q-type VGCC mutants, and found that there was no statistically significant difference in the amplitude (data not shown). (6) Stocks related to CICR component genes were *TRiP$^{JF03381}$* (dsRNA of *Rya-r44F*) from Bloomington Stock Center, and *Itp-r83A$^{90B.0}$* and *Itp-r83A$^{05616}$*(*Venkatesh and Hasan, 1997*) from Bloomington Stock Center. (7) Other transgenic lines were *UAS-dTrpA1.K (attP2)*, *UAS-Ca-α1D RNAi $^{JF01848}$*, *UAS-Ca-α 1D RNAi $^{HMS00294}$*, *UAS-white RNAi $^{GL00094}$*, *UAS-Lifeact:GFP*, *UAS-mito:HA:GFP*, *UAS-Kir2.1*, *UAS-GCaMP5G*, *GMR-Hid* and *UAS-Dcr-2* from Bloomington Stock Center, and *UAS-ChIEF-tdTomato$^{C3-3}$*(*Wang et al., 2011*) from Z Wang.

Exact genotypes of individual animals used in figures are described below:
*Figure 1*
(D) *3×[ppk-TN-XXL] (attP2)/ 3×[ppk-TN-XXL] (attP2)*
(E–G)*3×[ppk-TN-XXL] (attP2)/ 3×[ppk-TN-XXL] (attP2)* (WT)
*3×[ppk-TN-XXL] (attP40)/+; dTrpA1$^1$/ dTrpA1$^1$*
*pain$^3$/pain$^3$; 3×[ppk-TN-XXL] (attP2)/+*
*Figure 2*
(A–C,F) *3×[ppk-TN-XXL] (attP40)/+*
(D) *3×[ppk-TN-XXL] (attP40)/+* (WT)
*3×[ppk-TN-XXL] (attP40)/+; ppk-Gal4/UAS-Kir2.1*
*Ca-α1D$^{X10/AR66}$; 3×[ppk-TN-XXL] (attP2)/+*
*Ca-α1D$^{X7/AR66}$; 3×[ppk-TN-XXL] (attP2)/+*
(E) *Ca-α1D$^{X10/AR66}$; 3×[ppk-TN-XXL] (attP2)/+*
*Figure 3*
(A,B,D–J) *3×[ppk-TN-XXL] (attP2)/ 3×[ppk-TN-XXL] (attP2)*
*Figure 4*
(A–D,F) *w$^{1118}$/+; ppk-Gal4/+; 3×[ppk-TN-XXL] (attP2)/+* (Control)
*ppk-Gal4/+; 3×[ppk-TN-XXL] (attP2)/UAS-white RNAi $^{GL00094}$*
*ppk-Gal4/+; 3×[ppk-TN-XXL] (attP2)/UAS-Ca-α1D RNAi $^{JF01848}$*
*ppk-Gal4/+; 3×[ppk-TN-XXL] (attP2)/UAS-Ca-α1D RNAi $^{HMS00294}$*
(E) *w$^{1118}$/w$^{1118}$ or Y* (WT)
*Ca-α1D$^{X10/AR66}$*
(G) *ppk-Gal4/GMR-Hid; 3×[ppk-TN-XXL] (attP2)/+* (Control)
*ppk-Gal4/GMR-Hid; 3×[ppk-TN-XXL] (attP2)/UAS-Ca-α1D RNAi $^{JF01848}$*
*ppk-Gal4/GMR-Hid; 3×[ppk-TN-XXL] (attP2)/UAS-Ca-α1D RNAi $^{HMS00294}$*
*Figure 5*
(A–C) *UAS-ChIEF-tdTomato$^{C3-3}$/UAS-Dcr-2; ppk-Gal4/ UAS-Ca-α1D RNAi $^{HMS00294}$*

*Figure 6*
(A–E) 3×[ppk-TN-XXL] (attP2)/ 3×[ppk-TN-XXL] (attP2)

## Electrical resistance-based temperature measurements

To measure the temperature of microenvironment, we employed electrical resistance thermometry using a glass microelectrode (*Palmer and Williams, 1974*; *Shapiro et al., 2012*; *Yao et al., 2009*). To elucidate the relationship between the electrical resistance of an electrode and the temperature of the external saline solution (*Xiang et al., 2010*), we monitored the electrical resistances at various values of temperature of the solution. First, we heated the saline solution in a petridish up to 50°C with an inline heater (SF-28, Warner Instruments, Hamden, CT) that was regulated by a thermo-controller (TC-324B, Warner Instruments). Then, we turned off the heater and recorded the electrical resistances ($R$) of the electrode and the temperatures ($T$) of the solution simultaneously during natural cooling. The electrical resistances of glass microelectrodes were 5–10 MΩ at ambient temperature (~25°C). We measured the electrical resistances by giving square pulses (50 ms, 10 mV; 1 Hz) in the voltage clamp mode. The reciprocal of temperature ($1/T$) was plotted against the log of the electrical resistance ($log\ R$), so that the Arrhenius equations were estimated as a $R$-$T$ transformation formula by linear regression.

## Electrophysiology, IR-laser irradiation, and short-wavelength light stimulation

Fillets were made from wandering third instar larvae with the cuticle facing down in the external saline solution (*Xiang et al., 2010*) (120 mM NaCl, 3 mM KCl,4 mM MgCl$_2$, 10 mM NaHCO$_3$, 10 mM trehalose, 10 mM glucose, 5 mM TES, 10 mM sucrose, 10 mM HEPES. pH adjusted to 7.25 with NaOH). To immobilize larvae, all of the segmental nerves were cut, and then central nervous systems were removed. Muscles covering the neurons of interest were gently digested by infusion of Protease Type XIV (0.5% w/v, Sigma-Aldrich, St. Louis, MO) through a glass micropipette (~1.2 MΩ) mounted on a PatchStar micromanipulator (Scientifica, East Sussex, UK). After the enzymatic treatment, a couple of perfusions of external saline solution were carried out to remove excess enzyme, hemocytes, and other cellular debris. No detectable difference in dendrite morphology of Class IV dendritic arborization neurons was observed before and after muscle digestion, indicating the neurons were intact.

Class IV neurons were identified by the fluorescence of TN-XXL driven by the *ppk* promoter (see details in Ca$^{2+}$ imaging). Recording pipettes were pulled with a P-1000 puller (Sutter Instruments, Novato, CA) from thin wall borosilicate glass (B150-110-10, Sutter Instruments), filled with external saline solution, with a tip opening of 5 μm (800 kΩ–1MΩ). Gentle negative pressure was delivered to suction the soma to get good signal-to-noise ratios of recording traces. We mostly recorded activity of v'ada of Class IV neurons. Recordings were performed with a Multiclamp 700B amplifier (Molecular Devices, Sunnyvale, CA), and data were acquired with Digidata 1440A (Molecular Devices) and Clampex 10.0 software (Molecular Devices). Extracellular single-unit recordings of action potentials were obtained in current clamp mode, with an 8 kHz low-pass filter and sampled at 10 kHz or 20 kHz.

Data were analyzed by Clampfit 10.3.1.5 software (Molecular Devices) and custom programs written in MATLAB (MathWorks, Natick, MA). Extracellular spikes were detected by first band-pass filtering (the high-pass filter: RC monopole, $f_c$ = 100 Hz; the low-pass filter: Butterworth 8-pole, $f_c$ = 2 kHz) and then thresholding the extracellular voltage trace with the 'Threshold Search' plugin of Clampfit software. The Maximum firing rates were computed by sliding a rectangular window function along the spike train with $\Delta t$ = 400 ms. The peaks of the time derivative of firing rate fluctuation were assigned as follows: first, the spike density estimation was computed by using the Gaussian kernel method ($\sigma$ = 25 ms; *Shimazaki and Shinomoto, 2010*); second, the temporal differences of firing rate were calculated (time derivative of firing rate fluctuation; time interval = 0.1 ms); third, a threshold was configured as 0.5 of the primal peak value; finally, the peaks of the time derivative above the threshold were specified.

'Time to Max. Firing Rate' was defined as the left edge of the sliding rectangular time window that gave the earliest maximum firing rate in each spike train. To perform simultaneous recording from the soma and the axon bundle, we first obtained the extracellular recording from the soma,

and then the cutting edge of the axon bundle in the same hemisegment was suctioned into the recording electrode with a resistance of 700–800 kΩ.

A 200-mW continuous wave oscillation IR-laser light source was connected to the microscope through the IR-LEGO-200 system (*Kamei et al., 2008*) (Sigma Koki, Tokyo, Japan) to provide IR-laser stimulation through a dry objective (40× UApo/340 NA 0.90, Olympus, Tokyo, Japan), and the IR-laser beam was focused onto the center of the field of view. The source of IR laser was a continuous wave semiconductor laser (SLD-1462-200-C; FiberLabs, Fujimino, Japan). Precise focus of the IR-laser was localized by the spotted heat evaporation of indelible marker ink (K-177N, Shachihata, Nagoya, Japan). All of the power values in the text and figures are settings of the laser driver. IR-laser power density was measured at the focal plane of the objective using a radiometric sensor head (UP17P-6S-H5-DO, Gentec-EO, Quebec, Canada) coupled with a power meter (UNO, Gentec-EO). The measured output power was ~48.2% of the preset power. Custom programs in Digidata 1440A controlled the duration and timing of IR-laser irradiation by opening a TTL-triggered shutter (Shutter Controller F77-6, Suruga Seiki, Shizuoka, Japan) in the IR-LEGO system.

A 100-W mercury light source (U-LH100HGAPO, Olympus) was connected to the microscope with a light guide to provide short-wave light stimulation through a filter (BP460-495, Olympus) and the objective, yielding an evenly illuminated light spot, which covered the entire Class IV dendritic arborization neuron. Light intensity was measured as in IR-laser irradiation. Custom programs in Digidata 1440A controlled the duration and timing of light illumination by opening a TTL-triggered shutter (SSH-R, Sigma Koki) in the light guide. To administrate 3 µM Tetrodotoxin (Nacalai, Kyoto, Japan) or 5 µM Nimodipine (Sigma-Aldrich), fillets were incubated in the chamber for 30 min before recording. 10 µM Thapsigargin (Nacalai) was employed to deplete the cytoplasmic $Ca^{2+}$ store.

## $Ca^{2+}$ imaging

$Ca^{2+}$ imaging of TN-XXL-expressing Class IV neurons was performed on fillet or whole-mount preparations. Fillet preparations were made essentially as in extracellular recording, except that the protease treatment was omitted when only $Ca^{2+}$ dynamics were recorded. For whole-mount imaging, larvae were anesthetized with isoflurane and mounted in halocarbon oil on a microscope slide. Data were collected on an IX-71 (Olympus) equipped with an objective (40× UApo/340 NA 0.90, Olympus), Nipkow disk confocal system (CSU-X1, Yokogawa Electric, Tokyo, Japan) and two EMCCD imagers (iXon X3 DU897-BV, Andor Technology, Belfast, UK). The custom-made camera mount (Olympus) of each EMCCD imager was equipped to adjust either the rotation angle around the optical axis (for CFP imaging) or the displacements along X- and Y-axes (for YFP imaging). Using the adjustment devices, the two imagers' fields of view were aligned with minimal distortion in the rotation angle and minimal displacements less than one pixel along the X- and Y-axes.

TN-XXL was excited with a 445-nm diode laser (CUBE 445-40C, Coherent, Santa Clara, CA). Images were acquired at 512×512 pixels or 256×256 pixels with 2x2 binning, in 14-bit dynamic range, and with 100 or 33 ms exposure time. CFP and YFP fluorescence signals were captured synchronously and separately by the two imagers with 10 or 30 Hz, through 490/40 and 578/105 band-pass filters (Semrock, Lake Forest, IL), respectively. In simultaneous acquisitions of extracellular single-unit recordings and $Ca^{2+}$ imaging, two imagers were controlled by Solis 4.19 software (Andor Technology) and the exposure timings were recorded by Digidata 1440A; these timing data were used for the offline processing as synchronization cues. In the case of the $Ca^{2+}$ image acquisition only, two imagers were controlled by iQ 1.10.3 software (Andor Technology). Data were analyzed by ImageJ (NIH) and custom programs written in MATLAB (MathWorks). The setting of regions of interest (ROI) was guided with the binary mask image generated by the dendritic morphology. The background signals were subtracted from CFP or YFP images; then the FRET ratios were calculated (see *Figure 2—figure supplement 1*). In response to IR stimulations, $Ca^{2+}$ fluctuations with $\Delta R_{peak}$ larger than 10% were designated as $Ca^{2+}$ transients.

## Behavioral assays

The thermal nociception behavioral tests were performed essentially as described (*Caldwell and Tracey, 2010*; *Hwang et al., 2007*), with slight modifications. Animals were raised at 25°C in an incubator with 12 hr light/dark cycles and humidity was manually controlled (75–80%). Third instar larvae

were gently picked up from the vial, washed twice with deionized water, and transferred to a 140×100-mm petridish with fresh 2% agarose. Excessive water was removed from the animals. For acclimation, animals were allowed to rest on the plate for at least 5 min before testing. The response latency was measured as the time interval from the point at which the larva was first contacted by the probe until it completed the first 360° roll.

The light nociception behavioral tests were performed essentially as described (*Xiang et al., 2010*), with slight modifications. Animals were raised and treated as in the thermal nociception test. The assay was carried out with a Macro Zoom Microscope system (MVX-10, Olympus). Light was delivered from a 100-W mercury lamp (U-LH100HGAPO, Olympus) through a MV PLAPO 1X objective (Olympus) at 2.5X magnification, yielding a light spot of 1.5 mm in diameter. Other devices we used were a shutter (SSH-R1X, Sigma Koki) in the light guide, a filter for the background light (30.5S-R64, Olympus), and a lens for a CCD camera (GE60, Library, Tokyo, Japan). A filter (GFP-3035D, Olympus) was placed into the above microscope to illuminate blue light (472.5 nm with 30 nm band width). The light intensity at 2.5X magnification was measured by a radiometric sensor head (UP17P-6S-H5-DO, Gentec-EO, Quebec, Canada) coupled with a power meter (UNO, Gentec-EO) and it was determined to be 0.72 mW/mm$^2$ in the light spot. Sixty-five to 104 animals were tested in each condition and the percentage of positive responses was calculated.

## Optogenetic neural activation

Larvae for optogenetic activation experiments, harboring the *UAS-ChIEF-tdTomato* transgene (*Wang et al., 2011*), were grown in the dark at 25°C for 4 days on fly food containing all trans-retinal (R2500; Sigma-Aldrich) at a final concentration of 0.5 mM. For behavioral experiments, one larva at a time was separated from food and placed into the center of a 9.5 x 13.5 cm square plastic dish filled with 2% agar (Bacto Agar; Becton Dickinson, Franclin Lakes, NJ). After 5 min of acclimation, the dish with larva was placed into the behavior rig. 5 cycles of 100 ms pulses followed by 100 ms pause intervals or a single 1 s long pulse of blue light (0.234 mW/mm$^2$; *Figure 5B*), or a single 1 s long pulse (0.056 mW/mm$^2$ for the crawling assay below) were applied by using a collimated LED light source (M470L3-C1; ThorLabs, Newton, NJ) with an emission peak at around 470 nm. A data acquisition system (USB-6212 BNC; National Instruments, Austin, TX) provided the triggering TTL signals, with timing controlled by a custom LabVIEW program (National Instruments). We recorded videos of larval behavior by using a GE60 CCD image sensor (640 x 480 pixels; Library) at 30 frames per second. For the analyses of crawling stride speeds, larval locomotions were traced in videos by using 'Manual Tracking' Fiji plugin (ImageJ 1.46J, NIH, Bethesda, MD), and then the maximum stride speeds for 5 s before and after optogenetic activation were calculated. For electrophysiological recordings, 5 cycles of 100 ms pulses with 100 ms pause intervals or a single 1 s long pulse of blue light (0.294 mW/mm$^2$ for *Figure 5A*), or a single 1 s long pulse (0.056 mW/mm$^2$) was applied from above through external saline solution. The Digidata 1440A digitizer (Molecular Devices) generated the triggering TTL signals, with timing controlled by a custom Clampex protocol (Molecular Devices).

## Quantitative PCR

RNA was purified from four larval CNS complexes for each genotype using an RNeasy kit (QIAGEN, Hilden, Germany). ReverTra Ace and Thunderbird (TOYOBO, Osaka, Japan) were used according to the manufacturers' instructions. Sequences of the primers were: 5'-GCT TGT GCC CAG GGA GC-3' and 5'-AGG ACA CAA TGT CCG GAT GAT CG-3' (Probe 1), and 5'-GCG ACC AGA ATG GCG ACT TTA ATG-3' and 5'-CCC GTA TCC ACT GGG ATG GAC-3' (Probe 2) for *dTrpA1*; 5'-GCG ACA CCC AAG TTA TTA AGG GTC-3' and 5'-GTT CAT CAA ACG TTG GCA GAT GC-3' (Probe 1), and 5'-TGC TGA CAG GCG AGT TTG AC-3' and 5'-GCC TGA GCC TTA ATA ACT GGG GTG-3' (Probe 2) for *pain*; and 5'-GCT AAG CTG TCG CAC AAA TG-3' and 5'-GTT CGA TCC GTA ACC GAT GT-3' for *rp49* used as a reference. Data were analyzed using the comparative $C_T$ method on a StepOne Real-Time PCR System (Applied Biosystems, Foster City, CA).

## Acknowledgements

The reagents, genomic datasets, and/or facilities were provided by the *Drosophila* Genetic Resource Center at Kyoto Institute of Technology, the Bloomington Stock Center, Vienna *Drosophila* Resource

Center, the TRiP at Harvard Medical School (NIH/NIGMS R01-GM084947), the NIG stock center, the Developmental Studies Hybridoma Bank at the University of Iowa, the *Drosophila* Genomics Resource Center (DGRC), FlyBase, modENCODE, YN Jan, T Sakai, P Garrity, C Duch, RW Ordway, JT Littleton, S Ryglewski, Z Wang, T Suzuki, M Miura, T Kanamori, K Emoto, and S Yonehara. We also thank Y Xiang, N Tanaka, K Horikawa, T Ishihara, T Teramoto, S Takagi, M Suzuki, Y Kamei and Y Tanaka for advice on electrophysiology, IR-mediated heating and $Ca^{2+}$ imaging; J Hejna for polishing the manuscript; and K Oki, J Mizukoshi, and M Futamata for their technical assistance. This work was supported by a grant from the programs Grants-in-Aid for Scientific Research on Innovative Areas 'Mesoscopic Neurocircuitry' (22115006 to T Uemura and 22115005 to MM), a grant from Takeda Science Foundation to T. Uemura, a Grant-in-Aid for Scientific Research (C) to T Usui (24500410), a grant from the programs Grants-in-Aid for Scientific Research on Innovative Areas 'Brain Environment' to T Usui (24111525), and the Platform Project for Supporting in Drug Discovery and Life Science Research (Platform for Dynamic Approaches to Living System) from the Ministry of Education, Culture, Sports, Science (MEXT) and Japan Agency for Medical Research and development (AMED). S-IT, DM and KO are recipients of a JSPS Research Fellowship for Young Scientists.

## Additional information

### Funding

| Funder | Grant reference number | Author |
| --- | --- | --- |
| Ministry of Education, Culture, Sports, Science, and Technology | Grants-in-Aid for Scientific Research on Innovative Areas Mesoscopic Neurocircuitry, 22115006 | Tadashi Uemura |
| Takeda Science Foundation | | Tadashi Uemura |
| Japan Society for the Promotion of Science | Research Fellowship for Young Scientists | Shin-Ichiro Terada Daisuke Matsubara Koun Onodera |
| Ministry of Education, Culture, Sports, Science, and Technology | Grant-in-Aid for Scientific Research (C), 24500410 | Tadao Usui |
| Ministry of Education, Culture, Sports, Science, and Technology | Grants-in-Aid for Scientific Research on Innovative Areas Brain Environment, 24111525 | Tadao Usui |
| Ministry of Education, Culture, Sports, Science, and Technology | Platform Project for Supporting in Drug Discovery and Life Science Research of AMED | Tadashi Uemura |
| Ministry of Education, Culture, Sports, Science, and Technology | Grants-in-Aid for Scientific Research on Innovative Areas Mesoscopic Neurocircuitry, 22115005 | Masanori Matsuzaki |

The funders had no role in study design, data collection and interpretation, or the decision to submit the work for publication.

### Author contributions

S-IT, DM, KO, TUs, Conception and design, Acquisition of data, Analysis and interpretation of data, Drafting or revising the article, Contributed unpublished essential data or reagents; MM, Conception and design, Analysis and interpretation of data; TUe, Conception and design, Analysis and interpretation of data, Drafting or revising the article

### Author ORCIDs

Tadashi Uemura, http://orcid.org/0000-0001-7204-3606
Tadao Usui, http://orcid.org/0000-0002-0507-1495

## Additional files

**Supplementary files**

• Source code 1. The code to select regions of interest (ROIs), measure the intensities of both CFP and YFP in each ROI, and compute the FRET ratios.

• Source code 2. The code to manipulate the timings of LED illumination and the imaging in the optogenetics behavioral assay.

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
