## [Decision Letter]

Thank you for submitting your work entitled "Neuronal processing of noxious thermal stimuli mediated by dendritic Ca^2+^ influx in *Drosophila* somatosensory neurons" for consideration by *eLife*. Your article has been favourably evaluated by K VijayRaghavan (Senior Editor) and three peer reviewers, one of whom, Leslie Griffith, is a member of our Board of Reviewing Editors.

The reviewers have discussed the reviews with one another and the Reviewing Editor has drafted this decision to help you prepare a revised submission.

Summary:

This study describes the response of Class IV sensory neurons in *Drosophila* larvae to distinct noxious stimuli (heat and light). Using a highly sophisticated experimental setup combining UV-mediated focal heat application, electrophysiology and calcium imaging, the authors show that "burst and pause" activity pattern of Class IV neurons accompanying a calcium transient is induced by heat but not light. In *dTrpA1* mutants or L-type voltage-gated calcium channel mutants, the "burst and pause" activity pattern, calcium transients and the behavioral response to noxious heat (rolling) are blocked. Optogenetic induction of an activity pattern mimicking the "burst and pause" activity slightly increased the frequency of rolling. From these results, the authors propose an interesting hypothesis that specific pattern of neural activity mediates transmission of specific modality.

Essential revisions:

The one concern was that the linkage between burst coding and rolling is not as strong as the prose indicates. The authors should reword their conclusions in the Abstract, Results and Discussion in more conservative language and make it clear that rolling can (and does) occur without bursting. Bursting appears to enhance rolling, but it is not necessary. In particular, the phrase "necessary and sufficient" needs to be removed since the data do not support this claim.

---

## [Author Response]

Essential revisions: The one concern was that the linkage between burst coding and rolling is not as strong as the prose indicates. The authors should reword their conclusions in the Abstract, Results and Discussion in more conservative language and make it clear that rolling can (and does) occur without bursting. Bursting appears to enhance rolling, but it is not necessary. In particular, the phrase "necessary and sufficient" needs to be removed since the data do not support this claim.

We now realize that it is appropriate to reword our conclusions in more conservative language in order to make it clear that rolling occurs without bursting. Also, we fully agree with the notion that bursting appears to enhance rolling, but it is not essential. In our revised manuscript, we rephrased our conclusions in more modest and cautious manner. In particular, we removed the phrase "necessary and sufficient”. All the revised phrases or sentences are positioned in the following sections in the revised version: Abstract; Introduction, last paragraph; Results, subsection “Optogenetic experiments strongly suggest that multiple fluctuations in firing rate in Class IV neurons enhances robust rolling behavior” heading and second paragraph; Discussion, subsection “Roles of Ca^2+^ spikes in nociception of Class IV neurons”, third paragraph.